

# Determining the link between hygroscopicity and composition for semi-volatile aerosol species

Joel Alroe[1], Luke T. Cravigan[1], Mark D. Mallet[1], Zoran D. Ristovski[1], Branka Miljevic[1], and Graham R. Johnson[1]

[1]School of Chemistry, Physics and Mechanical Engineering, Queensland University of Technology, Queensland, Brisbane, 4001

*Correspondence to*: Zoran D. Ristovski (z.ristovski@qut.edu.au)

**Abstract.** Internally and externally mixed aerosols present significant challenges in assessing the hygroscopicity of each aerosol component. This study presents a new sampling technique which uses differences in volatility to separate mixtures and directly examine their respective composition and hygroscopic contribution. A shared thermodenuder and unheated bypass line are continuously cycled between an aerosol mass spectrometer and a volatility and hygroscopicity tandem differential mobility analyser, allowing real-time comparative analysis of heated and unheated aerosol properties. Measurements have been taken of both chamber-generated secondary organic aerosol and coastal marine aerosol at Cape Grim, Australia, to investigate system performance under diverse conditions. Despite rapidly changing aerosol properties and the need to restrict analysis to a narrow size-range, the former experiment separated the hygroscopic influences of ammonium sulfate and two distinct organic components with similar oxygen to carbon ratios but different volatilities. Analysis of the marine aerosol revealed an external mixture of non-sea salt sulfates and sea spray aerosol, both of which likely shared similar volatile fractions composed of sulfuric acid and a non-hygroscopic organic component.

## 1 Introduction

Atmospheric aerosols have important roles in air quality and the climate. These roles are strongly affected by their hygroscopicity, which represents the capacity of aerosol to adsorb water vapour from the surrounding air. Amongst other effects, this water uptake promotes aqueous chemistry and can lead to the formation of cloud droplets.

Organic aerosol (OA) compounds make a large contribution to global aerosol mass (Jimenez et al., 2009), and they include a huge range of compounds with varying chemical and hygroscopic properties. Furthermore, many are volatile, transitioning between the particle and gas phase in response to changes in their gas-phase concentration and local atmospheric conditions (Seinfeld and Pankow, 2003; Donahue et al., 2012). These volatile compounds are often more reactive, subject to oxidative and oligomeric processes, and their dynamic nature complicates predictions of bulk aerosol properties.

Extensive lab-based and atmospheric studies of organic species have reported a wide range of organic hygroscopicities. For example, secondary organic aerosols (SOA) condensed from the oxidation products of α-pinene have demonstrated hygroscopic growth factors ranging from 1.01 to 1.4 (Prenni et al., 2007; Duplissy et al., 2008). This range is due to the



mixture of oxidation products generated by varying concentrations of precursors, oxidisers, environmental conditions, aging processes and experiment duration.

Volatility-based methods can directly separate aerosol components and have been used to investigate their independent hygroscopic contributions. Most commonly, this is achieved by passing the aerosol sample through a heated thermodenuder

(TD), causing a volatile component to desorb, and examining the resulting change in properties. Until recently, this technique has been separately applied to either measurements of composition or hygroscopicity (Hong et al., 2014; Johnson et al., 2004; Sellegri et al., 2008; Villani et al., 2013; Huffman et al., 2008). Three recent studies have combined these measurements by splitting heated samples from a TD between an aerosol mass spectrometer (AMS) and a cloud condensation nuclei (CCN) counter (Cain and Pandis, 2017; Cerully et al., 2015; Hildebrandt Ruiz et al., 2015).

Cain and Pandis (2017) and Hildebrandt Ruiz et al. (2015) examined chamber-generated SOA from α-pinene and toluene respectively, while Cerully et al. (2015) performed atmospheric measurements of rural aerosol from the south-eastern United States. Across these three cases, a clear relationship between hygroscopicity and oxidation level (often measured as an oxygen to carbon ratio, O:C) could not be established. In addition, the least volatile organic compounds were shown to have the lowest hygroscopicity, which contrasts with many previous studies. Cain and Pandis (2017) found that low volatility

compounds had typically high O:C ratios, while Hildebrandt Ruiz et al. (2015) observed conflicting trends. Cerully et al. (2015) did not report either trend for O:C ratio but found correlation between low volatility and high average carbon oxidation state ($\overline{OS_C}$), an alternate measure of oxidation level (Kroll et al., 2011). In short, each study highlighted the complexity of organic hygroscopicity and the importance of volatility-based methods in distinguishing their competing contributions.

The above techniques for measuring bulk aerosol properties are convenient to implement, however generally they do not examine size-dependent variations in hygroscopicity or the presence of external mixing. Although water uptake is typically more strongly influenced by particle size than by composition (Dusek et al., 2006), the role of composition becomes highly significant near the critical diameter for cloud droplet activation and a large proportion of the aerosol number concentration is often centred there (Wex et al., 2010). The critical diameter is sensitive to changes in aerosol hygroscopicity, so the

composition of aerosol at or near the critical diameter can significantly affect the number concentration of available CCN (Andreae and Rosenfeld, 2008; Mallet et al., 2017). Similarly, bulk analysis of externally mixed aerosol can be confounded by the competing hygroscopic influences of aerosol from different sources. Measurement techniques which separate these mixtures can assist with source apportionment and investigation of each aerosol type.

In this article, we describe a novel real-time sampling system which couples an AMS to a volatility and hygroscopicity

tandem differential mobility analyser (VH-TDMA), and discuss how this combined approach provides valuable insight into the properties of the semi-volatile component.



## 2 Methodology

### 2.1 Sampling system design

The AMS provides real-time non-refractory chemical speciation and is capable of size-dependent measurements derived from the particle time of flight (PTOF). A full explanation of its design is given by Drewnick et al. (2005). For this study, a

compact Time of Flight AMS has been used which generates spectra with unit mass resolution and offers maximum sensitivity for aerosol with vacuum aerodynamic diameters ($d_{va}$) between 100 – 600 nm (Liu et al., 2007; Takegawa et al., 2009).

The VH-TDMA measures aerosol volatility and hygroscopicity under sub-saturated conditions. The function of our instrument has been explained in other papers (Johnson et al., 2005; Fletcher et al., 2007). In short, aerosol of a pre-selected

narrow size range is heated in a compact TD with maximum temperature of 500 °C and residence time of 3 s. The sample then passes to two scanning mobility particle sizers (labelled: V-SMPS and H-SMPS, respectively), with a humidified stage preceding the H-SMPS. These measure the mobility diameter ($d_m$) change due to loss of volatile material and subsequent hygroscopic growth, respectively. Particle hygroscopicity is reported as a hygroscopic growth factor (HGF), representing the relative increase in particle diameter after humidification compared to the dry diameter.

To obtain simultaneous hygroscopic and compositional measurements, the two instruments have been coupled together through a combined flow system, shown in Fig. 1. After passing through a shared inlet, the aerosol is dried to a relative humidity (RH) of approximately 30 % using a membrane dryer (Nafion MD-700). If additional drying is required, a silica gel diffusion dryer is also added. To assist electrical mobility-based measurements in the VH-TDMA, a well-defined charge distribution is applied to the sample with a Kr-85 neutraliser (TSI Model 3012A). The RH and temperature of this dried

aerosol flow is measured with a humidity probe (Rotronic HC2-WIN-USB) which is recessed in a sealed T-junction to avoid disrupting the sample flow. Since air exchange in the recessed region is diffusion limited, the probe provides an estimate of mean inlet RH, rather than precise time-resolved measurements.

The flow is then evenly split between the TD and an unheated bypass line (labelled: Line A and Line B, respectively). A series of fast-acting solenoid valves are used to direct these two sampling flows so that one instrument measures heated

aerosol while the other receives the unheated sample. Under normal sampling conditions, the flow paths are switched between the instruments every three minutes, allowing them both to perform a continual series of consecutive heated and unheated measurements. Changes in HGF, composition and particle diameter between these alternating samples can be attributed to the removal of the semi-volatile species. The switching process is automated using control software developed in LabView. This software also manages the VH-TDMA pre-selected aerosol diameter, TD temperature and humidifier

settings. These can be progressively stepped through a range of set values to investigate size dependencies, volatility distributions or to reveal any deliquescence or efflorescence transitions between dry and aqueous particle phases.

The two instruments sample at different flowrates. To avoid flowrate-dependent variations in transmission efficiencies and temperature fluctuations in the TD, flowrates of 1 L min⁻¹ are maintained in both sampling lines. Since this is lower than the



combined flowrate required for the V- and H-SMPS, mass flow controllers provide supplementary filtered air to these components and their measured particle concentrations are corrected for this dilution. Likewise, since the AMS requires only 0.1 L min$^{-1}$ sample flow, the additional 0.9 L min$^{-1}$ is maintained by a mass flow controller and vacuum supply connected near the AMS inlet. With the exception of a few short flexible connections using conductive silicon, stainless steel tubing

has been used throughout the sampling system, to avoid siloxane-based contamination of the hygroscopic and compositional measurements (Timko et al., 2009).

## 2.2 Size-resolved composition

In this study, all measurements were performed on aerosol with $d_m = 100$ nm as this is both sufficiently small for composition to have a significant influence on CCN-forming potential and large enough to be within the peak sensitivity

range of the AMS. If the aerosol is strongly size-dependent, the focus on ultrafine aerosol inhibits meaningful conclusions from bulk compositional analysis. In that case, more representative analysis can be obtained using PTOF size-resolved measurements, given with respect to $d_{va}$. The conversion to $d_{va}$ from $d_m$, using particle density ($\rho_p$), unit density ($\rho_0$) and the Jayne shape factor ($S$), can be calculated as follows (DeCarlo et al., 2004):

$$d_{va} = \frac{\rho_p}{\rho_0} S \times d_m \tag{1}$$

Since $\rho_p$ and $S$ are composition-dependent, PTOF mass concentrations are integrated over a $d_{va}$ range which best represents aerosol at the preselected $d_m$ used by the VH-TDMA. For heated measurements, this PTOF $d_{va}$ range is shifted downwards in proportion to the $d_m$ reduction observed with the VH-TDMA. AMS particle transmission efficiency decreases for diameters below $d_{va} = 100$ nm, reaching approximately 0 % transmission at 40 nm. Where PTOF data are required from this reduced sensitivity range, a linear correction factor is applied to account for transmission losses (Knote et al., 2011).

While this use of PTOF data provides composition that is more directly relevant to the VH-TDMA measurements, it encompasses a reduced fraction of the overall aerosol mass concentration and does not benefit from as much signal averaging as is available for bulk analysis. This is particularly exacerbated when high time resolution is required, or during periods of low mass loading, and can result in a highly variable timeseries. In these cases, a locally weighted smoothing method (LOESS) can be used to discern qualitative trends within the PTOF data.

## 2.3 Hygroscopic analysis

The hygroscopicity of small particles is reduced by the Kelvin effect, which causes water activity ($a_w$) at the droplet/air interface to increase with particle curvature. To account for this, the measured HGFs can be re-expressed in terms of the hygroscopicity parameter ($\kappa$), using $\kappa$-Köhler theory (Petters and Kreidenweis, 2007):

$$\frac{RH/100}{\exp\left(\frac{4\sigma M_w}{RT\rho_w D_d HGF}\right)} = \frac{HGF^3 - 1}{HGF^3 - (1-\kappa)} \, , \tag{2}$$



where RH is the relative humidity set in the H-SMPS, $\sigma$ is the droplet surface tension (assumed to be equivalent to pure water, $\sigma_w = 0.072$ J m$^{-2}$), $M_w$ is the molecular weight of water, $R$ is the universal gas constant, $T$ is the temperature, $\rho_w$ is the density of water and $D_d$ is the dry particle diameter. The $\kappa$ values can then be reverted to Kelvin-corrected HGFs ($HGF_{corr}$) by setting $a_w$ equal to RH/100:

$$\frac{1}{a_w} = 1 + \frac{\kappa}{HGF_{corr}^3} \qquad\qquad (3)$$

After excluding the effect of droplet curvature, the compositional influence on hygroscopicity can be investigated in detail. The hygroscopicity of internally mixed aerosol is commonly estimated using the following volume-weighted model (Stokes and Robinson, 1966):

$$HGF_{total}^3 = \sum_i \varepsilon_i HGF_i^3 \quad, \qquad\qquad (4)$$

where $\varepsilon_i$ and $HGF_i$ are the volume fraction and independent HGF contribution of each component. If the components have substantially different volatilities, these parameters can be directly determined from VH-TDMA measurements. Otherwise assumed HGF contributions are used and $\varepsilon_i$ is derived from the corresponding mass concentrations ($m_i$) and densities ($\rho_i$) of each component:

$$\varepsilon_i = \frac{m_i/\rho_i}{\sum_i m_i/\rho_i} \qquad\qquad (5)$$

## 2.4 Aerosol transmission efficiencies

The two sampling lines offer different transmission efficiencies due to variations in tubing geometry, an additional solenoid valve on the Line A, and losses associated with the TD itself. Diffusional losses in the TD have been reduced by omitting the cooling section. When sampling at high aerosol loading, this may cause recondensation of volatile species onto the aerosol as it cools at the outlet, however negligible recondensation is expected for most atmospheric samples (Saleh et al., 2011). The remaining differences in transmission efficiency have been quantified by examining size- and temperature-dependent changes in aerosol concentration between the two flow paths.

Ammonium sulfate (AS) aerosol was generated with a collision nebuliser. The aerosol was dried, neutralised and sampled at three preselected sizes: $d_m = 50$, 150 and 300 nm. The humidifier and H-SMPS were replaced with a condensation particle counter (CPC; TSI Model 3772) and a suitable bypass flow to maintain normal sampling flowrates throughout the system. From each sampling line, measurements were made with the TD at room temperature, to examine differences in tubing and solenoid valves. The TD temperature was then progressively increased in 5 °C increments up to the volatilisation point of AS (180 °C at 50 nm in this system), to quantify thermophoretic losses.

Figure 2 displays the relative transmission efficiency of Line A compared to Line B. For diameters of 150 nm and above, path-dependent losses of less than 5 % were observed at room temperature. This increased to over 12 % at 50 nm. In addition, transmission efficiency decreased linearly with temperature, for all sizes. Based on these results, a constant



correction factor has been applied to all size-resolved AMS mass concentrations in this study, assuming a mean 85 % relative transmission efficiency for aerosol of $d_m \leq 100$ nm at 120 °C. This reduces bias between the two sampling lines and isolates the compositional changes caused by removal of volatile species.

## 2.5 Smog chamber sampling

A system test was conducted under controlled conditions to examine correlation between the two instruments and ensure that meaningful conclusions could be drawn. Measurements were performed using AS-seeded secondary organic aerosol (SOA), generated in a temperature-controlled, 8 m³ Teflon© smog chamber. A polydisperse distribution of AS seed particles were generated from a nebuliser, with an initial number concentration of $10^4$ cm⁻³ and geometric mean diameter of 94 nm. The chamber was humidified to 50 %RH and prepared with 35.5 ppb of α-pinene (Sigma-Aldrich). N-butanol (D9, 98 %, Sigma-
Aldrich) was added as a tracer, allowing OH concentrations to be monitored with a Chemical Ionisation Time-of-Flight Mass Spectrometer (Barmet et al., 2012). Gaseous nitrous acid (HONO) was injected into the chamber giving an initial OH concentration of $1.46 \times 10^7$ molecules cm⁻³. The chamber was irradiated by twenty 160 W UV lamps and sampling was conducted over a four-hour period.

VH-TDMA measurements were performed on 100 nm aerosol with a 6-minute time resolution (3 minutes per sampling
path). The TD was set to 120 °C, to target organic compounds with higher volatility than AS. The inlet RH was maintained at $32.6 \pm 0.3$ % throughout the experiment and H-SMPS humidifier was set to 90 %RH. Figure 3 gives an example of the direct measurements obtained from both instruments, from the first 2.5 hours of alternating heated and unheated measurements.

## 2.6 Remote coastal measurements

To investigate the suitability of the combined system for atmospheric sampling, it was deployed to the Cape Grim Baseline Air Pollution Station for a two-week measurement campaign in March 2015. This remote site on the northwest coast of Tasmania, Australia, frequently receives strong westerly winds carrying marine aerosol with negligible terrestrial or anthropogenic influences. From 22:00 on 2ⁿᵈ March, these baseline conditions were observed for an 8-hour period, with mean particle number concentrations of $400 \pm 60$ cm⁻³ for aerosol diameters above 10 nm. This period was accompanied by
significantly increased sulfate-related mass concentration and a pronounced bimodal particle number size distribution (Fig. 4), consistent with cloud-processed marine aerosol (Hoppel et al., 1986). The final two hours of baseline sampling was accompanied by a pronounced decrease in sulfate- and ammonium-related signal and a decrease in particle number concentration, suggesting a change in the air mass and associated aerosol properties. For this reason, analysis has been focused on the initial 6 hours which exhibited the most consistent properties.
To account for the low aerosol concentrations, all measurements have been averaged over this period, rather than examining trends over a higher time resolution. Likewise, uncertainties have been determined from standard deviations in the means.



The sampling inlet RH was consistently dried to below 30 % and sampling was conducted using the same pre-selected diameter, TD temperature and humidifier RH as in the chamber-based experiment (i.e. 100 nm, 120 °C, 90 % respectively).

## 3 Results and validation

### 3.1 Chamber-generated aerosol

Compositional measurements of the seeded SOA indicated a strong size dependence, with the AS mass distributed around a mean $d_{va}$ of approximately 500 nm, while the SOA component progressively dominated at 100-200 nm (Fig. 5a). Since the size-distribution of each species did not change substantially over time, compositional analysis was restricted to diameters in the range: $130 < d_{va} < 180$ nm. These diameter limits were selected to reflect varying proportions of internally mixed SOA and AS with densities of 1.3 and 1.78 g cm$^{-3}$, respectively (Chen and Hopke, 2009).

Detection limits for each species were calculated as three times the standard deviation of their background concentration, observed when sampling through a high-efficiency particle filter. Estimated detection limits were quite high due to the use of high time resolution samples, a narrow range of PTOF diameters, and the strongly time-dependent nature of the sample. Likewise, the observed signal was quite unstable, especially for heated measurements, often falling below the detection limit. For this reason, data below the detection limit was not removed as it would have excluded a large proportion of the data.

As expected, the $SO_4$ and $NH_4$ signals correlated strongly and were combined to give the total mass concentration of AS throughout the experiment. Similarly strong correlation was observed between the organic and $NO_3$ signals indicating the secondary formation of organonitrates or nitric acid from $NO_x$-based reactions. To simplify analysis, these species were summed to give the mass concentration of OA. Since there was no source of chloride in this experiment, and this species was constantly below the detection limit, it has not been included in the analysis.

As shown in Fig. 5b, during the first 100 minutes of the experiment, a semi-volatile organic component rapidly formed, increasing particle volatility and reducing the HGF of the unheated aerosol from 1.5 to <1.1 (Fig. 3). After this point, there was no significant change to HGFs and volatility gradually decreased. Figure 5b also shows that the heated aerosol composition revealed an increasing proportion of less volatile organic compounds which did not desorb at 120 °C. For convenience, these two semi-volatile and less-volatile organic components are labelled as SVOA and LVOA, respectively. It

should be noted that there was a non-negligible concentration of organics at the beginning of the UV irradiation period. This can be attributed to dark reactions that occurred during the preceding 30 minutes while AS seed particles were being generated, after the injection of α-pinene, as supported by a progressive decrease in α-pinene concentrations during this time. The unit mass resolution parameterisation given by Canagaratna et al. (2015) was used to calculate elemental ratios of oxygen to carbon (O:C), averaged across all diameters with the assumption that these ratios would be size-invariant. The OA

concentrations were most stable and well-resolved during the final 30 minutes of the experiment. Based on this period, both the LVOA and SVOA components exhibited similar O:C ratios of 0.43 ± 0.16 and 0.45 ± 0.16, respectively.



To confirm the predictive power of this combined sampling approach, the measured HGFs of the heated sample were compared against the composition-based model defined in Eq. (4). Assumed densities of 1.3 g cm$^{-3}$ (Duplissy et al., 2008) and 1.78 g cm$^{-3}$ were used for α-pinene SOA and AS, respectively. AS calibration measurements indicated that the HGF of AS at 90 %RH was $1.58 \pm 0.03$, after correcting for residual water in the particle phase during pre-selection. The

hygroscopic contribution of the low-volatility SOA component was determined using the O:C parameterisation proposed by Massoli (2010), giving an $HGF_{LVOA}$ of $1.1 \pm 0.2$. This value agrees well with a range of α-pinene HGFs reported in other studies (Varutbangkul et al., 2006; Virkkula et al., 1999; Prenni et al., 2007; Cocker Iii et al., 2001). Applying these AS and LVOA parameters to the composition-based model produced HGFs which closely matched the direct VH-TDMA measurements of the heated sample, as seen in Fig. 6.

Having established agreement between the instruments, Eq (4) was adapted to determine the hygroscopic contribution of the SVOA using measurements of particle diameter ($d$) and HGF:

$$HGF_0^3 = \left(\frac{d_0^3 - d_{TD}^3}{d_0^3}\right) HGF_{SVOA}^3 + \left(\frac{d_{TD}}{d_0}\right)^3 HGF_{TD}^3 \ , \tag{6}$$

where the subscripts $0$ and $TD$ refer to the unheated and heated samples, respectively. This use of direct VH-TDMA measurements avoided any dependence on an assumed SVOA density. The first 20 minutes of the experiment was excluded

due to low SVOA concentrations which caused unreliable HGF estimates. The remaining data support a stable mean $HGF_{SVOA}$ of $1.02 \pm 0.02$ (Fig. 6). This low value mirrors other studies of semi-volatile organics (Meyer et al., 2009; Raatikainen et al., 2010), and is within the range of α-pinene HGFs observed by Prenni et al. (2007) and Denjean et al. (2015).

In the preceding steps of this analysis, the HGF contributions and volume fractions for all three components (AS, SVOA and

LVOA) were established. Using these parameters and the same additive model, the hygroscopicity of the unheated aerosol was calculated. The results closely agreed with the observed HGFs (Fig. 7), confirming that the derived parameters accurately describe the aerosol properties throughout the experiment.

## 3.2 Remote coastal measurements

During the initial 6 hours of sampling under baseline conditions at Cape Grim, mass concentrations were very low and there

was little evidence of size-dependent composition. To provide better statistics, the composition has been averaged across the full size range of the AMS. The non-refractory aerosol mass was dominated by sulfate (65.8 %), ammonium (12.9 %) and organic compounds (20.7 %) (Fig. 8), while the nitrate- and chloride-related signals were consistently below their respective detection limits. Comparison between the heated and unheated samples indicated that the organic fraction was entirely desorbed at 120 °C. This volatility was reflected in the O:C ratio of 0.24, which is consistent with semi-volatile oxygenated

OA observed in other AMS studies (Jimenez et al., 2009; Raatikainen et al., 2010). 42 % of the sulfates were desorbed, despite being well below the volatilisation temperature of ammonium sulfate. In contrast, the ammonium signal showed





minimal volatility, indicating that a significant proportion of the sulfate was in the form of sulfuric acid and ammonium bisulfate. This is further supported by an ammonium to sulfate molar ratio of $1.03 \pm 0.04$ in the unheated aerosol.

Humidification of the 100 nm pre-selected aerosol frequently produced a bimodal particle size distribution, indicating that the aerosol was externally mixed. The dominant proportion exhibited a mean HGF of $1.57 \pm 0.01$, while a second aerosol

type had a mean HGF of $1.90 \pm 0.04$. These growth factors are characteristic for non-sea salt (nss) sulfate-dominated particles and sea spray aerosol (SSA), respectively (Sellegri et al., 2008; Villani et al., 2013). In further support of this subdivision, the SSA aerosol comprised $7 \pm 2 \%$ of the observed number concentration of 100 nm particles, which is consistent with the total proportion of SSA reported by Quinn et al. (2017) at similar latitudes. There was no apparent difference in volatility at 120 °C, with a universal $12 \pm 2 \%$ reduction in particle volume; however, this increased their HGFs

to $1.61 \pm 0.02$ and $2.01 \pm 0.05$, respectively (Table 2).

The hygroscopic contribution of the semi-volatile component was estimated from its mean composition using Eq. (4), assuming a simple mixture of sulfuric acid and OA. The corresponding parameters are listed in Table 1, where the OA HGF has been determined from an O:C – HGF parameterisation (Massoli et al., 2010). This compositional model gave a semi-volatile HGF of $1.13 \pm 0.05$. Applying the same model to direct VH-TDMA measurements of the nss sulfate aerosol yielded,

a semi-volatile HGF of $1.2 \pm 0.3$, in agreement with the composition-based estimate.

Since both the SSA and nss sulfate aerosol shared similar volatilities, their semi-volatile components may have likewise have shared similar composition and hygroscopic contributions. Based on the VH-TDMA measurements of heated SSA HGF and the above semi-volatile HGF, the model predicted an unheated SSA HGF of $1.94 \pm 0.05$. This is in line with the observed value and strongly suggests that both aerosol types had accumulated similar semi-volatile sulfates and OA during their

atmospheric lifetimes.

## 4 Conclusions

A new sampling system has been developed which pairs a VH-TDMA and an AMS to obtain simultaneous measurements of hygroscopicity, volatility and composition. By cycling both instruments between heated and unheated sampling lines, properties of the semi-volatile fraction can be directly measured in near real-time, over a range of pre-selected diameters, TD

temperatures and humidities. Size- and temperature-dependent transmission efficiencies have been characterised for the two sampling lines and relative losses of up to 15 % are observed in the heated line at 120 °C (50 nm, AS).

Two diverse measurement campaigns demonstrated consistent agreement between the instruments, leading to reliable composition-based predictions of hygroscopicity. The chamber-based measurements involved rapidly changing aerosol and compositional analysis of a narrow range of aerosol diameters, while particle concentrations remained very low throughout

the marine campaign. As a result, low counting statistics introduced large uncertainties and inhibited more detailed compositional analysis. This may have influenced the poor agreement between hygroscopicity and O:C ratio, observed for



the SVOA component of α-pinene SOA. Alternatively, it may reflect that these parameters are not well correlated, as reported by Hildebrandt Ruiz et al. (2015).

The capacity of both instruments to perform size-resolved measurements allowed size-dependencies to be identified and targeted in the analysis of the chamber-based campaign. Furthermore, this combined system is well suited to analysing complex mixed aerosols. The distribution of HGFs in the marine aerosol revealed an external mixture of nss sulfates and SSA. From these, an internally mixed semi-volatile fraction was separated and attributed to sulfuric acid and an OA component with a low degree of oxidation (O:C = 0.24). Finally, hygroscopic modelling supported the assumption that this semi-volatile component was common to both aerosol types.

In summary, these findings demonstrate the value of pairing both instruments through a shared TD to directly link the composition of volatility-separated components to their hygroscopic contribution. It is hoped that future measurements using this method will help improve models of aerosol-cloud interactions, particularly in regions subject to high levels of volatile organic emissions or mixed aerosols from diverse sources.

## 5 Data availability

The underlying research data can be accessed upon request to the corresponding author (Zoran Ristovski; z.ristovski@qut.edu.au).

## 6 Author contribution

Joel Alroe operated the instrumentation, analysed and interpreted the data and prepared the manuscript. Luke Cravigan assisted with operation of the VH-TDMA, and contributed to data analysis and interpretation. Marc Mallet assisted with operation of the VH-TDMA and AMS, and contributed to data interpretation and writing. Zoran Ristovski contributed to campaign organization, data interpretation, writing and supervised the work of Joel Alroe. Branka Miljevic contributed to campaign organization, data interpretation, writing and supervised the work of Joel Alroe. Graham Johnson contributed to data interpretation, writing and supervised the work of Joel Alroe.

## 7 Acknowledgements

This work was funded by an Australian Government Research Training Program Scholarship and an ARC Discovery grant (DP150101649). The authors would like to acknowledge and thank the Cape Grim Baseline Air Pollution Station staff for providing on-site support and meteorological data, and Alastair Williams from the Australian Nuclear Science and Technology Organisation for associated radon-222 measurements.



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





**Figure 1: Schematics of the VH-TDMA / AMS sampling system, demonstrating the alternating flow paths used for consecutive samples.**





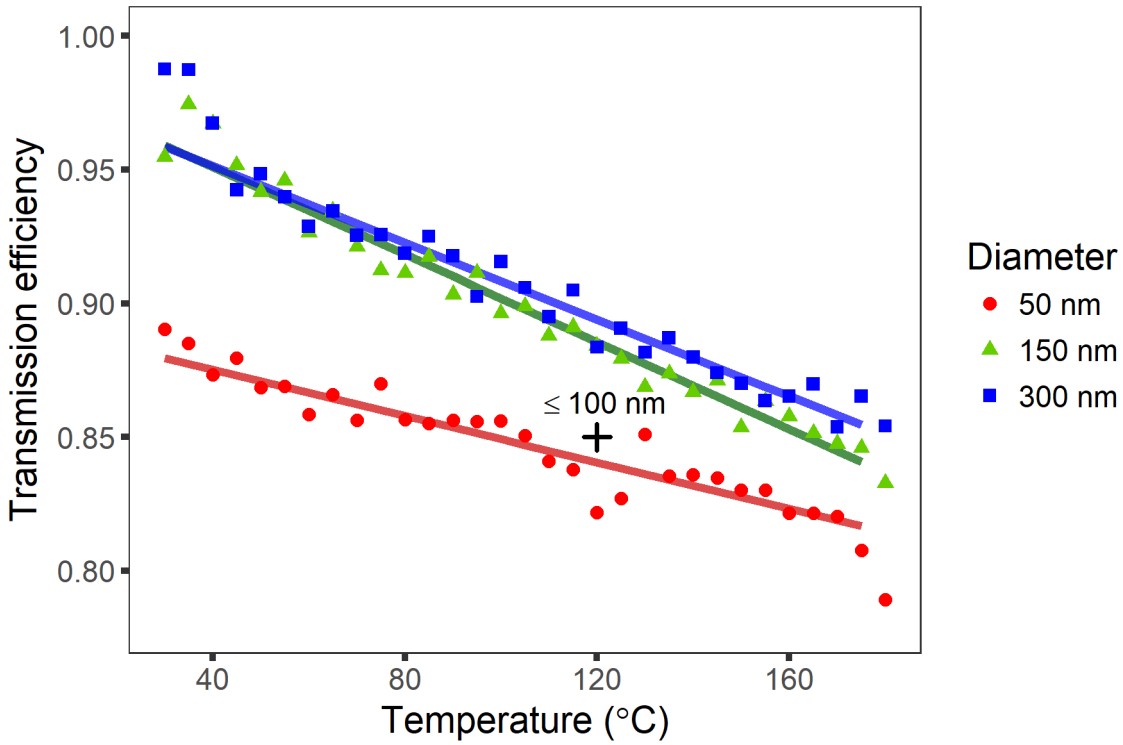

**Figure 2: Size- and temperature-dependent aerosol transmission rates of the heated sampling line compared to the unheated line, for AS aerosol. The value used in this study for aerosol ≤100 nm has been marked for comparison.**







**Figure 3: Consecutive measurements of unheated and heated aerosol properties during the first 2.5 hours of α-pinene SOA formation on AS seed particles. Gaps in the HGF data are due to faulty H-SMPS measurements which have been discarded.**





**Figure 4: (a) Mass concentrations of major compositional species and (b) size distributions of particle number concentration observed at the Cape Grim site on 2-3 March 2015. Dashed lines delimit a 6-hour period of relatively stable baseline sampling conditions.**




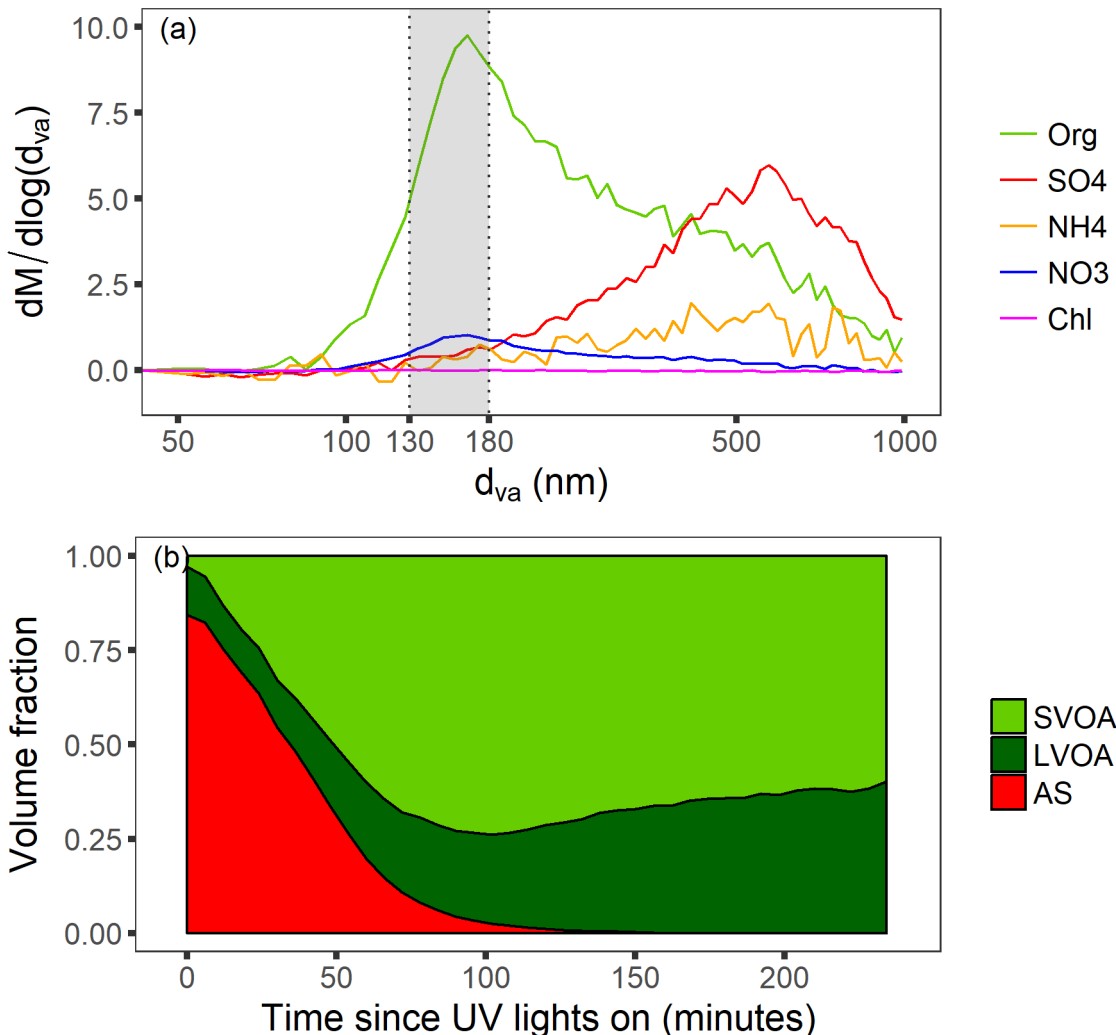

**Figure 5: (a) Mean size distribution of the major compositional species measured by the AMS, observed over four hours of SOA condensation and ripening. The shaded region represents the approximate $d_{va}$ range for particles with $d_m = 100$ nm. (b) The relative volume contributions of AS, LVOA and SVOA for aerosol in the selected $d_{va}$ range.**





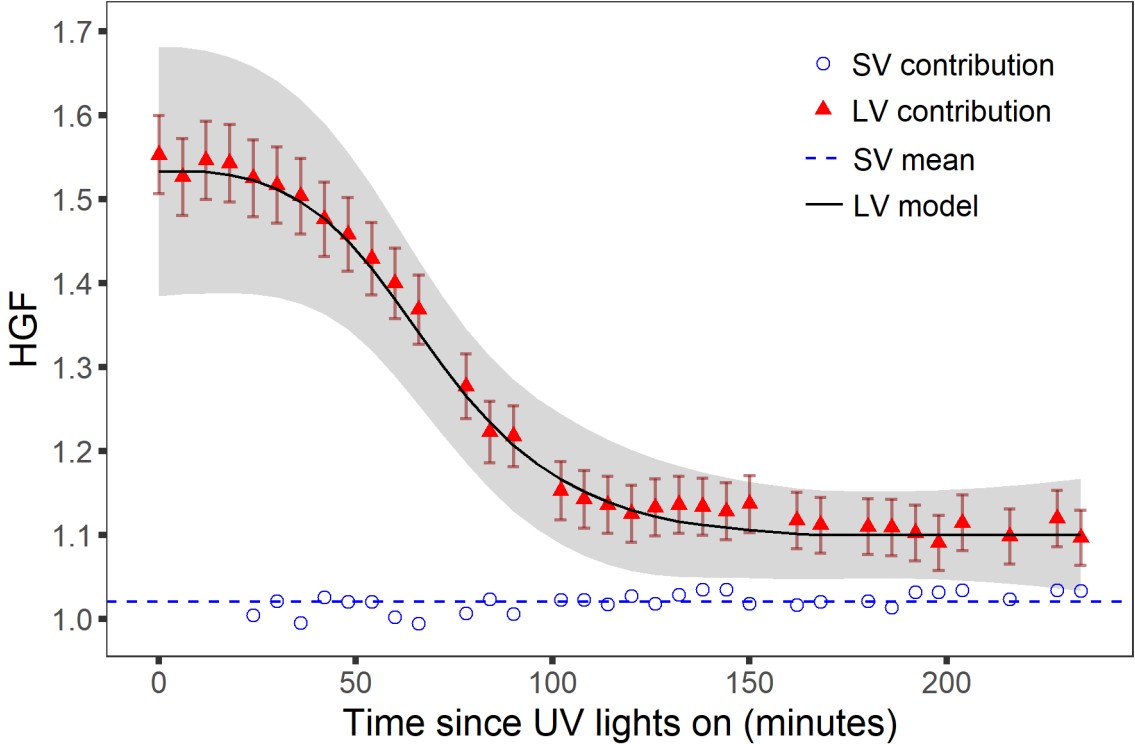

**Figure 6: Comparison between measured (data points) and modelled (smooth curve) HGFs for the lower volatility component during SOA formation. Error bars represent ±2 % uncertainty in H-SMPS measurements, while the shaded region gives a 99 % confidence interval for this model. The derived HGF of the semi-volatile component is shown for contrast.**





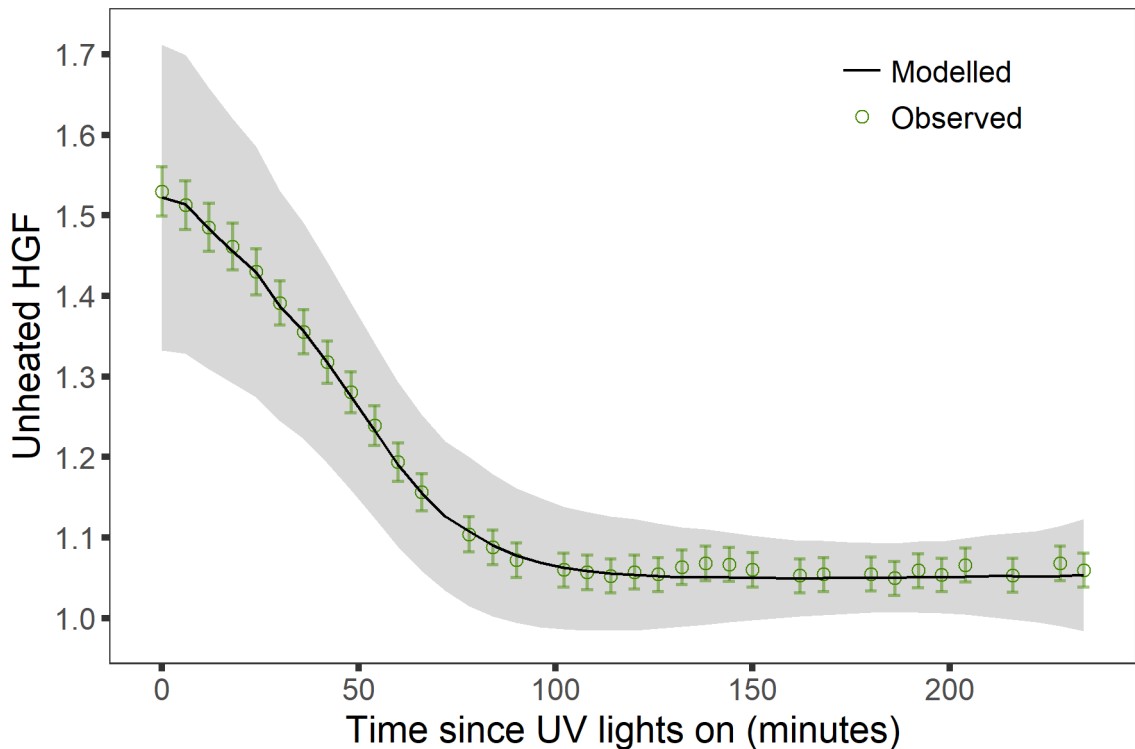

**Figure 7: Observed and predicted hygroscopicity of unheated, internally mixed α-pinene SOA and AS aerosol during four hours of SOA formation. Error bars represent ±2 % uncertainty in H-SMPS measurements, while the shaded region gives a 99 % confidence interval for this model.**



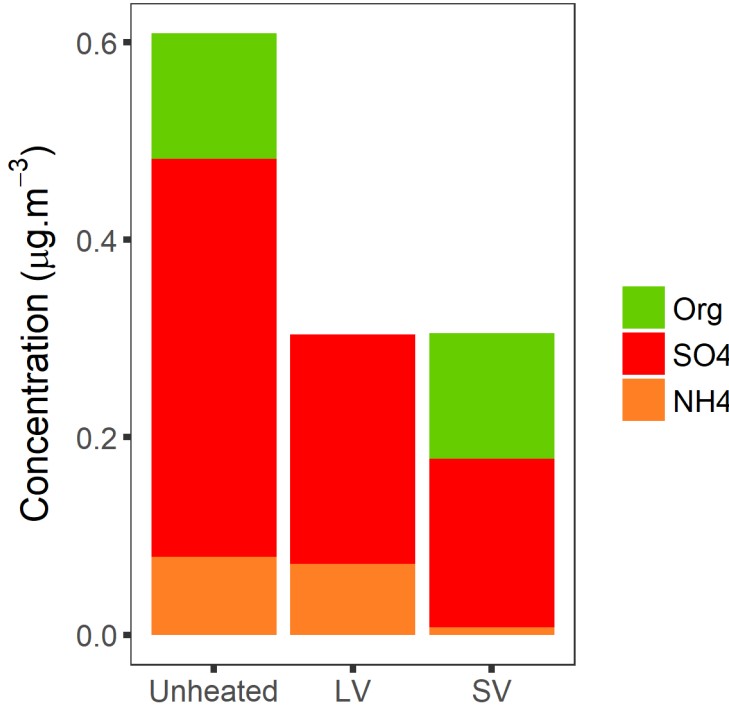

**Figure 8: Contribution of sulfates, ammonium and non-refractory organic compounds to the total, semi-volatile and low-volatility fractions of marine aerosol at Cape Grim.**



**Table 1: Parameters used in the compositional model of marine semi-volatile hygroscopicity**

| Component | Density (g cm$^{-3}$) | HGF contribution |
|---|---|---|
| OA | $1.2 \pm 0.1^a$ | $1.0 \pm 0.1$ |
| Sulfuric acid | $1.83 \pm 0.01^b$ | $1.95 \pm 0.05^c$ |

**[a] (Ault et al., 2013; Hersey et al., 2009)**

**[b] (Washburn, 1926)**

**[c] (Xiong et al., 1998)**

**Table 2: Hygroscopicity of volatility-resolved fractions observed in externally mixed marine aerosol**

| Classification | Semi-volatile volume fraction | Semi-volatile HGF | Heated HGF | Total unheated HGF |
|---|---|---|---|---|
| nss sulfate | $0.12 \pm 0.02$ | $1.2 \pm 0.3^a$ | $1.61 \pm 0.02$ | $1.57 \pm 0.01$ |
| SSA | $0.12 \pm 0.02$ | $1.2 \pm 0.3^a$ | $2.01 \pm 0.05$ | $1.90 \pm 0.04$ |

**[a] Derived from VH-TDMA measurements of nss sulfate aerosol**