# Peer review of "Determining the link between hygroscopicity and composition for semi-volatile aerosol species"

_Atmospheric Measurement Techniques, 2018_

## Referee Comment (RC2)

Review of: "Determining the link between hygroscopicity and composition for semi-volatile aerosol species" by Alroe et al., submitted to *Atmospheric Measurement Techniques*

This manuscript (amt-2018-17) reports an approach that couples the Aerosol Mass Spectrometer (AMS) to a volatility and hygroscopicity tandem differential mobility analyser (VH-TDMA) setup. This approach allows separation of the semi-volatile and low volatility components and comparison to chemical composition. The main novel advance of this approach over other similar approaches is the incorporation of size dependent aerosol chemical composition from the AMS, which allows investigation of aerosol chemical composition in the size range most relevant to the VH-TDMA experiments and to cloud droplet activation.

The manuscript is well written and within the scope of *Atmospheric Measurement Techniques*. The manuscript may be publishable if the below major comments associated with data quality and the use of the thermodenuder are addressed in revision.

Major Comments:

1. The first major comment relates to data quality and echoes many of the comments from Anonymous Reviewer 1. The authors assert that the main advantage of their approach over previous similar approaches is the incorporation of size dependent aerosol chemical composition measurements. However, the authors also state that, for the chamber measurements, which had a much higher mass concentration than the ambient measurements, "the observed signal was quite unstable…often falling below the detection limit. For this reason, data below the detection limit was not removed as it would have excluded a large proportion of the data" (page 7, lines 13-15). If the signal is so unstable and a meaningful measurement so difficult to obtain, it is unclear how this approach represents an advance over previous versions, which is the key argument of the paper. The robustness of this approach must be discussed in substantial detail in any revision. Included in that discussion must be details concerning the experiments (e.g. whether the AMS measurements in Fig. 3 represent the fraction around 100 nm in the smog chamber and, if not, what the mass concentrations around 100 nm were during that experiment; what time resolution was used in the AMS measurements and how that compares to the time dependent composition changes; what is, and what factors are governing, the AMS limit of detection; what mass concentrations are required for this approach to be viable; etc.).

2. The second major comment relates to the use of the thermodenuder approach. It is known that many chemical components of secondary organic aerosol (e.g. oligomers) can thermally decompose when passed through a thermodenuder at temperatures as low as 100ºC (Hall and Johnston, *Aerosol Sci. Technol.*, **2012**, *46*, 983-989). This observation may have a significant impact on the interpretation of the VHTDMA measurements, especially since the thermodenuder used in this manuscript is ramped up to 500ºC. In the revised manuscript, the authors should include a discussion of the limitations of the thermodenuder approach with respect to separation of semi-volatile and low volatility components against likely changes to aerosol chemical composition resulting from thermal decomposition within the thermodenuder.

Additional Comments:

1.  In their revised manuscript, the authors need to better clarify the temperature threshold that separates semi-volatile from low volatility. Is the cut-off at 120ºC? This is inferred in the text (page 7, lines 23-25) but is not stated in a clear and direct manner. The authors should more clearly define what is meant (functionally) by SVOC and LVOC.

2.  Page 4, line 10: Do the authors mean "If the aerosol _chemical composition_ is strongly size-dependent…."?

3.  The authors should ensure all references are accurate. For example, Cerully et al. (2017) and Huldebrandt Ruiz et al. (2015) were both published in _Atmos. Chem. Phys._ but their references indicate _Atmos. Meas. Tech._ as journal in which they were published.

---

## Referee Comment (RC1) · Anonymous Referee #1 · 27 Feb 2018

**Overview:**

Alroe et al. describe a method for measuring the hygroscopicity of aerosols which separates the contributions of semi-volatile and low volatility components. Their approach combines measurements from an Aerosol Mass Spectrometer (AMS) and a Volatility and Hygroscopicity Differential Mobility Analyser (VH-TDMA), which together provide information about composition and hygroscopicity. The use of a thermodenuder (TD) allows the comparative impact of semi-volatile aerosol species to be assessed.

Overall the manuscript is interesting and within the scope of AMT. My major concern relates to data quality: while the extension compared to previous work is that the technique provides size-resolved (around 100 nm) and time-resolved data, the composition measurements do not appear sensitive enough to make this feasible in practice. Since all ambient and a large proportion of laboratory AMS measurements are reported to be below detection limit at this size range, it is not clear under what circumstances the full approach described is actually applicable to atmospheric measurements, or lab experiments. The authors therefore need to better justify their methodology and clarify the limitations of averaging over size bins and smoothing. I have some additional queries related to the analysis methods and overall clarity of the experiment descriptions which should be addressed before the manuscript is considered for publication.

**General comments:**

**Experimental description:** Many more details are required as to how the chamber experiments were performed (section 2.5): For instance: How much n-butanol was added? How was RH introduced, and what purity of water was used? How was the HONO prepared and introduced? Was the chamber mixed? How was the chamber cleaned? What were the background concentrations (particles, AMS, CIMS...) prior to the experiment?

**Data quality:** The authors admit in section 3.1 (P7 L10-14) that using highly timeresolved and size-resolved AMS data result in "unstable" data "often falling below the detection limit", even for lab experiments. If Fig 3 is a typical time series for the size range studied (130-180 nm), these concentrations seem more than sufficient for confidence in the AMS data. What is the source of this instability, then, and what is the size and origin of the background signal used to estimate the LOD? Can individual error bars be marked on in Fig 3? Without a more detailed discussion here it is difficult to be confident that the technique is viable and the results presented are meaningful in terms of quoted uncertainties.

**Smog chamber experiments** – **aerosol dynamics:** The authors note the rapidly changing conditions in the smog chamber experiments. However, substantially more information is required to understand the time series presented (Fig 3, 5-7). For instance, it is not clear how the aerosol mass loading ( $\mu$ g/m3) and size distribution

 $(dN/dlogD_p)$  actually evolve through the experiment – please present these as a function of time. It looks from Fig 5(a) that there are multiple maxima in the mass distribution – is this also the case in the number distribution, and if so why for a seeded experiment?

Related to this, one of the main driving forces for the rapidly decreasing HGF over time at 100nm is a decrease in sulfate, which is not even mentioned in the text. What are the causes of this? Can particle coagulation and wall loss rates be quantified, for instance?

I cannot reconcile the composition time series in Fig 3 and 5b. For instance, at 150 minutes in Fig 3, the sulfate and LV organic mass concentrations are comparable, while in Fig 5b, virtually no sulfate is present. I am wondering if Fig 3 presents the total AMS concentration rather than the size-resolved data as implied. If so its inclusion should be justified and thoroughly clarified. It would be necessary to see the raw size-resolved data also plotted and discussed in the context of "Data quality", above.

**Smog chamber experiments – derivation of HGFs:** I have a number of queries about how the smog chamber hygroscopic growth factors were derived (P8 L1-22). Firstly, the ammonium sulfate HGF =  $1.58 \pm 0.03$  from calibration experiments is low compared a range of previous measurements and the E-AIM and AIOMFAC models (1.7-1.8) e.g. (Denjean et al., 2014; Lei et al., 2014). Please discuss this discrepancy and how it might propagate given that AS is the dominant hygroscopic component.

The contribution of LVOA is estimated based on a parameterisation of O:C vs HGF from Massoli et al. (2010). However, the authors cite recent work suggesting O:C may not be a good proxy for CCN activity and hygroscopicity. They also find the two OA components in their experiments have similar O:C. Why, then, was this parameterisation used? And more problematically, why only for the LVOA? The SVOA HGF was estimated via a residual approach, whereas the parameterisation would give the same HGF as LVOA.

What value for  $HGF_{LVOA}$  would be obtained using the same approach as SVOA, i.e. calculating a residual HGF in the heated sample after the sulfate contribution is accounted for? Would the model perform substantially less well with a single  $HGF_{OA}$ , as is used for the ambient samples? Given the combined uncertainties in  $HGF_{LVOA}$  and  $HGF_{SVOA}$  (which overlap), and different estimation methods, the conclusion that the two fractions have different HGF needs to be better supported.

**Specific comments:**

P1 L20-22: Please provide a general reference for this paragraph.

P1 L27: Clarify what is meant by "dynamic nature" – many volatile species are not particularly reactive or prone to condensation/evaporation. Perhaps indicate specific species of importance.

P3 L15: Replace "the two instruments" with "the two instruments (AMS and VH-TDMA)".

P3 L17: When is additional drying required? Was it required in any of this work? If not, delete.

P3 L26: The "Line A and B" terminology is confusing and is subsequently not used a great deal. According to Fig 1, A/B are not constant sampling lines but relate to the changing paths of the TD and unheated aerosol samples. Why not just use "TD" and "unheated" and remove "A/B" altogether? Similarly, for the wall-loss experiments (Fig 2), the exact path difference being monitored should be clarified.

P3 L30-31: Was this "stepping" performed here? If so, for which parameters?

P4 L1-2: What is the combined flowrate required, and hence the diluting flowrate? Given the interest in semivolatile partitioning, how was dilution of the gas phase accounted for?

P4 L8: "...all measurements were performed on aerosol with dm = 100 nm...". This may be the case for all VH-TDMA measurements, but this is in the composition section and the AMS sample is not pre-classified according to Fig 1. Size-dependent number concentrations (Fig 4) and composition (Fig 5a) data are also shown later. Please clarify.

P4 L10: "If the aerosol is strongly size dependent". Should this read "aerosol composition"?

P4 L18-19: Was this linear correction factor applied to any data here? If so, which?

P4 L20-24: Which data were smoothed in this study? Mention here and in corresponding Fig captions. Please define "LOESS".

P4 L26-P5 L5: I am surprised that a Kelvin correction is required for a dry diameter of 100 nm. What was the magnitude of the correction?

P5 L30: Please change to "...transmission efficiency decreased linearly with increasing temperature...".

P6 L12: Justify precision of [OH] – what is the uncertainty?

P6 L16: The mention of Fig 3 here, without discussion, confuses section 3.1 of the results. I suggest this Figure is not mentioned until it is discussed in the results.

P6 L30: Again, reference to "all measurements" is confusing here. For instance, Fig 4 shows time series, not 6-hour averages.

P7 L24: SVOA and LVOA are defined "for convenience" based on desorption at 120 °C. How does this threshold relate, approximately, to well-known measures of volatility, such as saturation concentration/vapour pressure, for the mass loadings used here?

P7 L26: Please provide a plausible mechanism or literature precedent for dark SOA production in these conditions. The method section also implies the seeds and RH were added prior to alpha pinene, rather than in the 30 minutes afterwards.

P9 L1: Could organosulfates also contribute to this volatile sulfate (and organic) signal?

P9 L9: The heated particles lost 12% of their volume, but apparently around 50% of their mass (Fig 8). Please explain this discrepancy.

**Technical comments:**

Fig 3: Please label the panels (a), (b) and (c) respectively and refer to them as such in the manuscript.

Fig 4 (b): The dynamic range of the colour scale tops out at  $\sim 10 \text{ cm}^{-3}$  to my eyes. Please adjust, or bin the data rather than using a continuous scale.

Fig 5 (a): Please add the units of dM/dlog(dva).

**References:**

Denjean, C., Formenti, P., Picquet-Varrault, B., Katrib, Y., Pangui, E., Zapf, P. and Doussin, J.-F.: A new experimental approach to study the hygroscopic and optical properties of aerosols : application to ammonium sulfate particles, Atmos. Meas. Tech., 7, 183–197, doi:10.5194/amt-7-183-2014, 2014.

Lei, T., Zuend, A., Wang, W. G., Zhang, Y. H. and Ge, M. F.: Hygroscopicity of organic compounds from biomass burning and their influence on the water uptake of mixed organic ammonium sulfate aerosols, Atmos. Chem. Phys., 14, 11165–11183, doi:10.5194/acp-14-11165-2014, 2014.

Massoli, P., Lambe, A. T., Ahern, A. T., Williams, L. R., Ehn, M., Mikkila, J., Canagaratna, M. R., Brune, W. H., Onasch, T. B., Jayne, J. T., Petaja, T., Kulmala, M., Laaksonen, A., Kolb, C. E., Davidovits, P. and Worsnop, D. R.: Relationship between aerosol oxidation level and hygroscopic properties of laboratory generated secondary organic aerosol (SOA) particles, Geophys. Res. Lett., 37, L24801, doi:10.1029/2010GL045258, 2010.

---

## Author Comment (AC1) · 1 Jun 2018

Response to Anonymous Reviewer 1 is uploaded as a supplement

Please also note the supplement to this comment:
https://www.atmos-meas-tech-discuss.net/amt-2018-17/amt-2018-17-AC1-supplement.zip
* * *

---

## Author Comment (AC2) · 1 Jun 2018

Response to Anonymous Reviewer 2 is uploaded as a supplement

Please also note the supplement to this comment:
https://www.atmos-meas-tech-discuss.net/amt-2018-17/amt-2018-17-AC2-supplement.zip
* * *

---

## Author Response (AR1)

**Referee #1**

*Alroe et al. describe a method for measuring the hygroscopicity of aerosols which separates the contributions of semi-volatile and low volatility components. Their approach combines measurements from an Aerosol Mass Spectrometer (AMS) and a Volatility and Hygroscopicity Differential Mobility Analyser (VH-TDMA), which together provide information about composition and hygroscopicity. The use of a thermodenuder (TD) allows the comparative impact of semi-volatile aerosol species to be assessed.*

*Overall the manuscript is interesting and within the scope of AMT. My major concern relates to data quality: while the extension compared to previous work is that the technique provides size-resolved (around 100 nm) and time-resolved data, the composition measurements do not appear sensitive enough to make this feasible in practice. Since all ambient and a large proportion of laboratory AMS measurements are reported to be below detection limit at this size range, it is not clear under what circumstances the full approach described is actually applicable to atmospheric measurements, or lab experiments. The authors therefore need to better justify their methodology and clarify the limitations of averaging over size bins and smoothing. I have some additional queries related to the analysis methods and overall clarity of the experiment descriptions which should be addressed before the manuscript is considered for publication.*

Author
The authors thank the reviewer for the detailed comments and suggestions that have helped us to refine the manuscript. We agree that the detection limits reported in the original version of the manuscript significantly limited the value of the laboratory-based results. After further examination of our approach, we have identified several changes to our analysis which have significantly improved these results. The manuscript has been updated with a detailed discussion of this revised analysis, particularly in Sections 2.3, 2.6 and 3.1.

In short, the averaging time for the compositional measurements has been increased which has improved both the detection limits and the stability of the signal for all species except $NH_4$. This has substantially improved the sensitivity of the measurement, raising a much greater proportion of the measurements above the detection limit. In addition, since the signal stability has improved, meaningful trends can be obtained without the need for any statistical smoothing. The detection limit for $NH_4$ remains high and the source of its variability has not been clearly identified, although it may be due to residual effects from high AS concentrations which had been sampled immediately prior to starting the filtered background measurements. A linear fit has been used to estimate the $NH_4$ concentrations, rather than discarding a significant source of AS mass. The resulting composition-based HGF models still obtain close agreement with the direct VH-TDMA measurements, so we are confident that this methodology offers meaningful results even when sampling rapidly changing, size-dependent aerosol.

The ambient marine analysis has not required substantial revision, since the concentrations of $NH_4$, $SO_4$ and organics were well above their detection limits, $NO_3$ is not relevant to baseline marine aerosol, and Chl is primarily present in refractory compounds which cannot be efficiently detected by the AMS. The non-refractory species which exhibited mass fractions

and volatility consistent with characteristic marine aerosol, and gave good agreement with the hygroscopic measurements obtained by the VH-TDMA.

In summary, now that the chamber experiment results have been revised, the sampling system has demonstrated internally consistent findings and significant utility under two quite challenging scenarios. Given that Cape Grim receives some of the cleanest air in the world, we anticipate that much more detailed analysis will be possible when sampling atmospheric aerosols in other locations, or during less size-dependent or rapidly evolving laboratory studies.

Responses to the additional queries have been included below. Please note that text coloured in red refers to the added text in the manuscript. All page and line numbers refer to the revised manuscript (Revised_Manuscript_TrackedChanges.docx), or supplementary material (Supplement.docx), where all changes have been tracked. If the text has been significantly changed, only the section number is given in this document (e.g. "Section 2.6").

**General comments**
*Referee's comment*
*1. Experimental description:*
*Many more details are required as to how the chamber experiments were performed (section 2.5): For instance: How much n-butanol was added? How was RH introduced, and what purity of water was used? How was the HONO prepared and introduced? Was the chamber mixed? How was the chamber cleaned? What were the background concentrations (particles, AMS, CIMS...) prior to the experiment?*

Author's answer
1. Section 2.6 has been extensively re-written to provide full detail about the chamber preparations and initial conditions of the experiment.

*Referee's comment*
*2. Data quality:*
*The authors admit in section 3.1 (P7 L10-14) that using highly time- resolved and size-resolved AMS data result in "unstable" data "often falling below the detection limit", even for lab experiments. If Fig 3 is a typical time series for the size range studied (130-180 nm), these concentrations seem more than sufficient for confidence in the AMS data. What is the source of this instability, then, and what is the size and origin of the background signal used to estimate the LOD? Can individual error bars be marked on in Fig 3? Without a more detailed discussion here it is difficult to be confident that the technique is viable and the results presented are meaningful in terms of quoted uncertainties.*

Author's answer
2. The following discussion of detection limits and measurement uncertainties has been added to Section 2.1:
P2 L31: "Detection limits for each species are calculated as three times the standard deviation of their background concentration, observed when sampling particle-free air through a high-efficiency particle filter (DeCarlo et al., 2006). This accounts for the instrument's background signal from stray ions and electronic noise. Uncertainties are given as the larger value of either the detection limit or the species-dependent measurement accuracy of the AMS. These accuracy estimates encompass uncertainties in the ionisation efficiencies, particle collection

efficiencies and the inlet flow rate and is commonly estimated as ± 37 % for organics, ± 35 % for SO$_4$ and Chl, and ± 33 % for NO$_3$ and NH$_4$ (Bahreini et al., 2008)."

The reviewer is correct that the bulk concentrations were more than adequate as seen in Fig 4a (previously Fig 3a). However, aerosol mass is strongly biased towards large diameter aerosol and aerosol in the desired size range ($130 < d_{va} < 180$ nm) represented only a small fraction of the total aerosol mass. In addition, during PTOF sampling, concentrations selected from this size range represent data from only a subset of the total sampling time. In short, the size-resolved measurements did not benefit from as much sampling time and signal averaging as the bulk measurements and, as a result, exhibited higher variability. In the original manuscript, both limitations were countered by smoothing the data with a non-parametric regression technique. After further examination, we have found that the compositional trends and detection limits are sufficiently improved by averaging to a time resolution of 12 minutes (6 minutes each of heated and unheated sampling).

The resulting 6-minute averaged PTOF measurements are shown in Fig S3 and uncertainties have been given as error bars. Detection limits were 0.123, 0.012, 0.023, 0.245 and 0.023 µg m$^{-3}$ for organics, NO$_3$, SO$_4$, NH$_4$ and Chl respectively. Section 3.1 has been substantially changed to include discussion of this revised analysis (P7 L9). In short, a large proportion of the measurements are now above these detection limits. The major exception was NH$_4$, which was highly variable and had a correspondingly high detection limit. Its concentrations have been estimated with a linear fit to the size-resolved PTOF measurements.

*Referee's comment*
*3. Smog chamber experiments – aerosol dynamics:*
*The authors note the rapidly changing conditions in the smog chamber experiments. However, substantially more information is required to understand the time series presented (Fig 3, 5-7). For instance, it is not clear how the aerosol mass loading (µg/m³) and size distribution (dN/dlogD$_p$) actually evolve through the experiment – please present these as a function of time. It looks from Fig 5(a) that there are multiple maxima in the mass distribution – is this also the case in the number distribution, and if so why for a seeded experiment?*

*Related to this, one of the main driving forces for the rapidly decreasing HGF over time at 100nm is a decrease in sulfate, which is not even mentioned in the text. What are the causes of this? Can particle coagulation and wall loss rates be quantified, for instance?*

*I cannot reconcile the composition time series in Fig 3 and 5b. For instance, at 150 minutes in Fig 3, the sulfate and LV organic mass concentrations are comparable, while in Fig 5b, virtually no sulfate is present. I am wondering if Fig 3 presents the total AMS concentration rather than the size-resolved data as implied. If so its inclusion should be justified and thoroughly clarified. It would be necessary to see the raw size-resolved data also plotted and discussed in the context of "Data quality", above.*

Author's answer
3. Two time series have been added to the supplement demonstrating the total non-refractory aerosol mass (Fig S1), and the number size distributions (Fig S2) observed throughout the chamber experiment. Since the chamber did not have a mixing fan, it likely took at least 30 minutes to become uniformly mixed and this passive diffusional mixing may explain the initial rapid decrease in total particle mass and SO$_4$ concentration. Approximately 50 minutes after

the lights were switched on, SOA was condensing at a sufficient rate to drive up the total particle mass (Figs 3, S3 and S1) and increase the mode diameter of the particle number size distributions (Fig S2). An additional paragraph has been added to the start of Section 3.1 commenting on these observations (P7 L3).

This chamber was constructed fairly recently and the chamber losses have not yet been fully characterised. Also, it is unclear when the chamber became uniformly mixed. So it is challenging to verify an appropriate initial reference concentration or loss rate for wall loss/coagulation corrections. However, when modelling composition-dependent HGFs, our calculations are based on the relative fractions of each species and so our results are not significantly dependent on loss-corrected particle concentrations.

As noted by the reviewer, there was a progressive decrease in AS volume fraction throughout the experiment (Fig 5b). Our analysis focused specifically on aerosol with $d_m = 100$ nm. Initially these were 100 nm AS seeds, but ongoing SOA formation allowed progressively smaller AS seeds to reach the 100 nm target size. As a result, the relative contribution of AS to 100 nm aerosol decreased over time, leading to a corresponding decrease in HGF.

The multiple maxima present in the mass distribution (Fig 5a), is likely due to two reasons. Firstly, compared to large diameter (high mass) AS seeds, only a relatively small quantity of condensed SOA is required to dominate the mass composition of small seeds. So the relative fraction of SOA is biased towards small sizes. In addition, the AMS mass distribution is presented in terms of $d_{va}$, which is density-dependent. So since the smallest particles have the greatest relative fraction of SOA, they likewise suffer the greatest reduction in density and $d_{va}$, causing separation between the AS- and SOA-dominant maxima in the mass distribution.

*Referee's comment*
*4. Smog chamber experiments – derivation of HGFs: I have a number of queries about how the smog chamber hygroscopic growth factors were derived (P8 L1-22). Firstly, the ammonium sulfate HGF = 1.58 ± 0.03 from calibration experiments is low compared a range of previous measurements and the E-AIM and AIOMFAC models (1.7-1.8) e.g. (Denjean et al., 2014; Lei et al., 2014). Please discuss this discrepancy and how it might propagate given that AS is the dominant hygroscopic component. The contribution of LVOA is estimated based on a parameterisation of O:C vs HGF from Massoli et al. (2010). However, the authors cite recent work suggesting O:C may not be a good proxy for CCN activity and hygroscopicity. They also find the two OA components in their experiments have similar O:C. Why, then, was this parameterisation used? And more problematically, why only for the LVOA? The SVOA HGF was estimated via a residual approach, whereas the parameterisation would give the same HGF as LVOA. What value for HGFLVOA would be obtained using the same approach as SVOA, i.e. calculating a residual HGF in the heated sample after the sulfate contribution is accounted for? Would the model perform substantially less well with a single HGFOA, as is used for the ambient samples? Given the combined uncertainties in HGFLVOA and HGFSVOA (which overlap), and different estimation methods, the conclusion that the two fractions have different HGF needs to be better supported.*

Author's answer
4. The authors would like to thank the reviewer for drawing attention to the low HGF used for AS in this experiment. After further investigation of the calibration measurements, we have found that the deliquescence point was reached at 82.0 ±0.5 %RH, rather than 80 %RH (Tang

1991), suggesting that the H-TDMA humidity was overestimated by 2 %RH in both the calibration and subsequent chamber-based measurements. The HGF of AS is strongly RH-dependent above its deliquescence point, so this could account for the discrepancy between the observed and published HGF of AS. Since the humidity dependence of α-pinene SOA HGFs is not well defined, it is not feasible to correct for this discrepancy. Therefore all chamber-based HGFs must be considered to represent water uptake at 88 %RH. The HTDMA humidity has been corrected in the manuscript (P6 L18 and P8 L24)

The reviewer has questioned our use of an O:C parameterization for HGF. While we share their concern about its reliability as reported in other studies, our aim was to obtain an independent estimate for the LVOA contribution to HGF. With this, the AMS measurements could be used to estimate an independent composition-based HGF for the heated aerosol which could be compared against the direct VH-TDMA measurements. If a contribution had been derived via the residual (ZSR) method, it would not have been possible to verify agreement between the two instruments. A published value could have been used instead, but a very wide range have been published and the O:C parameterization gave some basis for the chosen value. We do agree that both OA components have very similar O:C values, which suggests that the VH-TDMA-derived HGF of the semi-volatile component (1.02) could be applied to both organic fractions. However, near the end of the experiment, the HGF of the heated aerosol (1.11) is significantly higher than the VH-TDMA-derived semi-volatile HGF (Fig 6) despite the lack of a substantial AS fraction. Therefore, using the same ZSR-based HGF contribution for both OA components results in systematic underestimates of both heated and unheated HGFs.

**Specific comments**
*Referee's comment*
*1. P1 L20-22: Please provide a general reference for this paragraph.*

Author's answer
1. The following reference has now been provided (P1 L22):
Seinfeld, J. H., and Pandis, S. N.: Atmospheric Chemistry and Physics: From Air Pollution to Climate Change, 3 ed., John Wiley & Sons, Inc., New York, 2016.

*Referee's comment*
*2. P1 L27: Clarify what is meant by "dynamic nature" – many volatile species are not particularly reactive or prone to condensation/evaporation. Perhaps indicate specific species of importance.*

Author's answer
The authors agree that semi-volatile species are not necessarily "more reactive". The term "dynamic", was used to indicate that they more readily partition between particle and vapor phase due to changes in concentration and temperature. The associated sentences have been reworded as follows:
P1 L25: "Since many are semi-volatile, their relative partitioning between the particle and gas phase can be sensitive to concentration changes and local atmospheric conditions transitioning between the particle and gas phase in response to changes in their gas-phase concentration and local atmospheric conditions (Seinfeld and Pankow, 2003; Donahue et al., 2012). Furthermore, as their partitioning changes, they can become exposed to different phase-dependent chemical reactions. These volatile compounds are often more reactive, subject to oxidative and

oligomeric processes, and their These dynamic changes dynamic nature complicates predictions of bulk aerosol properties."

*Referee's comment*
*3. P3 L15: Replace "the two instruments" with "the two instruments (AMS and VH-TDMA)".*

Author's answer
3. The line has been replaced, as per the recommendation (P3 L6).

*Referee's comment*
*4. P3 L17: When is additional drying required? Was it required in any of this work? If not, delete.*

Author's answer
4. Additional drying is required when the ambient temperature and/or sample humidity is too high for the nafion dryer to achieve the desired 30 %RH inlet humidity (such as when sampling in tropical environments or from a nebulised aerosol source). Additional drying was not used in either of the studies discussed in this paper, however the diffusion dryer was used instead of the nafion dryer for the Cape Grim coastal measurements. To reflect this, the manuscript has been updated as follows:
P3 L8: "… the aerosol is dried to a relative humidity (RH) of approximately 30 % using a membrane dryer (Nafion MD-700) or a silica gel diffusion dryer."
P6 L16: "A nafion dryer was used to maintain the inlet RH at 32.6 ± 0.3 % throughout the experiment and the H-SMPS humidifier was set to 90 %RH."
P6 L37: "The sampling inlet RH was consistently dried with a diffusion dryer to below 30 % …"

*Referee's comment*
*5. P3 L26: The "Line A and B" terminology is confusing and is subsequently not used a great deal. According to Fig 1, A/B are not constant sampling lines but relate to the changing paths of the TD and unheated aerosol samples. Why not just use "TD" and "unheated" and remove "A/B" altogether? Similarly, for the wall-loss experiments (Fig 2), the exact path difference being monitored should be clarified.*

Author's answer
5. The authors agree with this recommendation and have updated the following lines in response:
P3 L14: Deleted the mention of Line A and B in parentheses
P5 L15: "… an additional solenoid valve on the TD line, …"
P5 L26: "Figure 2 displays the relative transmission efficiency of the TD line compared to the unheated line."
P9 L39: "…relative losses of up to 15 % are observed in the TD line at 120 °C (50 nm, AS)."
Fig 1: The labels "Line A" and "Line B" have been removed.
Fig 2: Caption has been updated, indicating that it depicts the "transmission rates of the TD sampling line compared to the unheated line"

*Referee's comment*
*6. P3 L30-31: Was this "stepping" performed here? If so, for which parameters?*

Author's answer
6. The RH and TD temperature remained constant throughout each of the studies discussed in this paper (i.e. no "stepping" performed). In the Cape Grim coastal study, the pre-selected diameter for the VH-TDMA was regularly cycled between 40, 100 and 150 nm. The diameter changes occurred after each pair of heated/unheated measurements. Results from the 40 and 150 nm samples have not been examined in this study as the focus was on aerosol large enough to be sampled by the AMS, but small enough to be close to the critical diameter for cloud droplet activation.

*Referee's comment*
*7. P4 L1-2: What is the combined flowrate required, and hence the diluting flowrate? Given the interest in semivolatile partitioning, how was dilution of the gas phase accounted for?*

Author's answer
7. Dilution does not significantly affect measurements for this system. There was no dilution in the AMS line. A vacuum supply ensured a continuous total sample flow of 1.0 L min$^{-1}$ through the AMS sampling path, from which the AMS sampled at a rate of 0.1 L min$^{-1}$. For the VH-TDMA, dilution occurred after all sizing was completed (after passing through the electrostatic classifiers) and directly before the aerosol passed into each condensation particle counter (CPC). The exact dilution ratios depend on the model of CPC and the desired aerosol flow rate within each SMPS. In the case of these the two studies, the H-SMPS had a dilution ratio of 1:1 (aerosol vs filtered dilution air) and no dilution was used for the V-SMPS and. For the aerosol species examined in this manuscript, any volatilisation within the CPC itself would be unlikely to reduce the particle diameter below the instrument's detection threshold of 10 nm.

To clarify that there was no dilution on the AMS side, the manuscript has been updated as follows (P3 L26): "Conversely, since the AMS requires only 0.1 L min$^{-1}$ sample flow, an additional 0.9 L min$^{-1}$ of sample flow is maintained by a mass flow controller and vacuum supply connected near the AMS inlet."

*Referee's comment*
*8. P4 L8: "...all measurements were performed on aerosol with $d_m$ = 100 nm…". This may be the case for all VH-TDMA measurements, but this is in the composition section and the AMS sample is not pre-classified according to Fig 1. Size-dependent number concentrations (Fig 4) and composition (Fig 5a) data are also shown later. Please clarify.*

Author's answer
8. The authors agree that this wording was unclear. Size-resolved particle time of flight (PTOF) AMS measurements were used in the chamber-based experiment to restrict compositional analysis to a comparable range of vacuum aerodynamic diameters. However the Cape Grim aerosol did not exhibit strongly size-dependent composition and no size-selection was applied to the AMS measurements for that experiment. Section 2.3 "Size-resolved composition" has been significantly re-worded to avoid implying that all AMS analysis was size-selected.

*Referee's comment*
*9. P4 L10: "If the aerosol is strongly size dependent". Should this read "aerosol composition"?*

Author's answer
9. This section (Section 2.3) has been significantly rewritten, and this comment has been incorporated into the new discussion.

*Referee's comment*
*10. P4 L18-19: Was this linear correction factor applied to any data here? If so, which?*

Author's answer
10. The linear correction factor was only applied to the chamber-based data. In that experiment, the composition was size-dependent and its analysis required size-resolved PTOF data including aerosol with diameters <100 nm. Since the coastal Cape Grim aerosol was not strongly size dependent, PTOF measurements were not used and therefore it was not necessary to apply any size dependent corrections to that dataset. To clarify this, the sentence regarding the linear correction factor has been reworded as follows:
P4 L20: "Since the chamber-based measurements, discussed in Section 3.1, required PTOF data from this diameter range, a linear correction factor was applied..."

*Referee's comment*
*11. P4 L20-24: Which data were smoothed in this study? Mention here and in corresponding Fig captions. Please define "LOESS".*

Author's answer
11. As discussed above, LOESS smoothing is not being applied to the chamber-based data now. Instead, the PTOF data from this experiment has been averaged to a 12 minute time resolution. When calculating AS concentrations for the HGF models, the $NH_4$ concentrations were approximated with a linear fit to the size-resolved $NH_4$ measurements. Section 3.1 has been significantly reworked to reflect these changes. No smoothing has been applied to the Cape Grim dataset.

*Referee's comment*
*12. P4 L26-P5 L5: I am surprised that a Kelvin correction is required for a dry diameter of 100 nm. What was the magnitude of the correction?*

Author's answer
12. The scale of the corrections are shown in the figure below. In short, the HGFs of the unheated chamber-generated aerosol increased by 0.6 – 3.5 % after the Kelvin correction. Diameters reduced by up to 30 % during heating, increasing the impact of the Kelvin correction and leading to HGF increases between 1.7 – 4.4 %. The largest corrections were observed during the first hour of the experiment because $\kappa$ is partially dependent on the uncorrected HGF (Equation 2, P4 L33). Since the range of corrections for both heated and unheated HGFs exceed the 2% measurement uncertainty of the H-TDMA, Kelvin-corrected HGFs have been used in this analysis.

[Figure]

*Referee's comment*
*13. P5 L30: Please change to "...transmission efficiency decreased linearly with increasing temperature...".*

Author's answer
13. The sentence has been updated as recommended (P5 L28): "…transmission efficiency decreased linearly with increasing temperature…"

*Referee's comment*
*14. P6 L12: Justify precision of [OH] – what is the uncertainty?*

Author's answer
14. The authors agree that the original OH concentration was overly precise. Barmet et al. (2012) reported an uncertainty of 25% in the rate constant which relates [butanol-d9] to [OH]. In light of this, the manuscript has been updated to indicate that this is only an estimate of [OH] (P6 L19): "initial OH concentration of approximately $1.5 \times 10^7$ molecules cm$^{-3}$"

*Referee's comment*
*15. P6 L16: The mention of Fig 3 here, without discussion, confuses section 3.1 of the results. I suggest this Figure is not mentioned until it is discussed in the results.*

Author's answer
15. This sentence has been reworded and moved to Section 3.1 (P6, L4). The figure numbering has also been updated, and the corresponding figure is now Fig 4.

*Referee's comment*
*16. P6 L30: Again, reference to "all measurements" is confusing here. For instance, Fig 4 shows time series, not 6-hour averages.*

Author's answer

16. Figure 4 uses a higher time resolution to demonstrate broader trends in composition and number size distribution on 2-3 March 2015. It is presented prior to any discussion of data analysis and its purpose is to support the choice of a restricted 6 hour time period for averaging and further detailed examination. To clarify this, the reference to "all measurements" has been reworded as follows:

P6 L34: "To account for the low aerosol concentrations, aerosol properties were averaged over this 6 hour period and the resulting mean values were used for all further analysis."

*Referee's comment*
*17. P7 L24: SVOA and LVOA are defined "for convenience" based on desorption at 120°C. How does this threshold relate, approximately, to well-known measures of volatility, such as saturation concentration/vapour pressure, for the mass loadings used here?*

Author's answer

17. The authors acknowledge the value of reporting volatility in terms of equilibrium saturation concentrations, or the volatility basis set. However the thermodenuder used in this study has a short residence time of approximately 3 seconds, which is insufficient for the aerosol to reach equilibrium (Riipinen et al., 2010). In addition, the seeded α-pinene experiment was not repeated for different SOA loadings and only one thermodenuder temperature was used throughout the experiment. Under these circumstances, it is our understanding that it is non-trivial to determine meaningful equilibrium saturation concentrations and is outside the scope of this study. However, volatility basis set analysis will certainly be a valuable addition for future studies using this methodology

*Referee's comment*
*18. P7 L26: Please provide a plausible mechanism or literature precedent for dark SOA production in these conditions. The method section also implies the seeds and RH were added prior to alpha pinene, rather than in the 30 minutes afterwards.*

Author's answer

18. The authors have not found literature supporting similar dark SOA formation. However in light of the improved analysis discussed above, it seems that there was negligible organics present at the start of the experiment (Figs 5b and S3).

The smog chamber method section (Section 2.6) has been re-ordered to reflect the chronological sequence of events, shifting the sentence regarding injection of AS seeds (P6 L12) to directly before UV illumination of the chamber (P6 L13).

*Referee's comment*
*19. P9 L1: Could organosulfates also contribute to this volatile sulfate (and organic) signal?*

Author's answer

19. It is possible that organosulfates (OS) contributed to the sulfate mass fraction observed at Cape Grim, especially since OS formation is promoted by acidic aerosol (Surratt et al., 2007). Their fragmentation pattern within a unit mass resolution AMS is largely indistinguishable

from inorganic sulfates (Farmer et al., 2010), so it is not possible to conclusively identify an OS fraction in this Cape Grim dataset. However since OS compounds often have low volatility (Lukács et al., 2009; Liggio and Li, 2006), and other marine studies have reported relatively low OS contributions to the total organic mass (Hawkins and Russell, 2010; Claeys et al., 2010), it is unlikely that they contribute significantly to the volatile component in this study.

*Referee's comment*
*20. P9 L9: The heated particles lost 12% of their volume, but apparently around 50% of their mass (Fig 8). Please explain this discrepancy.*

Author's answer
20. The AMS relies on flash vaporization of aerosol at 600 °C and does not efficiently detect refractory compounds, such as sea salt. Therefore, while 50 % of the non-refractory mass was desorbed in the TD, it is likely that this comprised only 12 % of the total aerosol volume. The remaining 88 % of the aerosol volume was likely composed of refractory compounds (including sea salt), and non-refractory compounds which were not fully desorbed at 120 °C. To clarify this, P9 L6 has been reworded as follows:
49.5% of this non-refractory mass was desorbed at 120 °C, including the entire non-refractory organic fraction.
And the following sentence has been added to P9 L21:
The difference between the volume and mass fractions which remained after heating imply the presence of a substantial volume of refractory material (such as SSA) which could not be efficiently detected by the AMS.

**Technical comments**
*Referee's comment*
*1. Fig 3: Please label the panels (a), (b) and (c) respectively and refer to them as such in the manuscript.*

Author's answer
1. This figure has been renumbered as Fig 4, and the panels have been labelled as suggested.

*Referee's comment*
*2. Fig 4 (b): The dynamic range of the colour scale tops out at ~10 cm$^{-3}$ to my eyes. Please adjust, or bin the data rather than using a continuous scale.*

Author's answer
2. This figure has been renumbered as Fig 3. Panel (b) has been replotted with an improved colour scale and expressing concentrations in terms of $dN/dlog(d_m)$. The y-axis has also been expanded to display the full diameter range measured by the SMPS (5 – 200 nm).

*Referee's comment*
*3. Fig 5 (a): Please add the units of dM/dlog(dva).*

Author's answer

3. The y-axis label of Fig 5 (a) has been updated with appropriate units as follows: "dM/dlog($d_{va}$) ($\mu g\ m^{-3}$)"

**Referee #2**

*This manuscript (amt-2018-17) reports an approach that couples the Aerosol Mass Spectrometer (AMS) to a volatility and hygroscopicity tandem differential mobility analyser (VH-TDMA) setup. This approach allows separation of the semi-volatile and low volatility components and comparison to chemical composition. The main novel advance of this approach over other similar approaches is the incorporation of size dependent aerosol chemical composition from the AMS, which allows investigation of aerosol chemical composition in the size range most relevant to the VH-TDMA experiments and to cloud droplet activation. The manuscript is well written and within the scope of Atmospheric Measurement Techniques. The manuscript may be publishable if the below major comments associated with data quality and the use of the thermodenuder are addressed in revision.*

Author

The authors appreciate the reviewer's comments that have helped us to refine the manuscript. We have implemented several changes to our analysis which have significantly improved the quality of the data and subsequent derived findings. The manuscript has been updated with a detailed discussion of this revised analysis, particularly in Sections 2.3, 2.6 and 3.1.

Responses to the reviewer's comments have been included below. Please note that text coloured in red refers to the added text in the manuscript. All page and line numbers refer to the revised manuscript (Revised_Manuscript_TrackedChanges.docx) and supplementary material (Supplement.docx) where all changes have been tracked. If the text has been significantly changed, only the section number is given in this document (e.g. "Section 2.5").

**Major comments**

*Referee's comment*

*1. The first major comment relates to data quality and echoes many of the comments from Anonymous Reviewer 1. The authors assert that the main advantage of their approach over previous similar approaches is the incorporation of size dependent aerosol chemical composition measurements. However, the authors also state that, for the chamber measurements, which had a much higher mass concentration than the ambient measurements, "the observed signal was quite unstable…often falling below the detection limit. For this reason, data below the detection limit was not removed as it would have excluded a large proportion of the data" (page 7, lines 13-15). If the signal is so unstable and a meaningful measurement so difficult to obtain, it is unclear how this approach represents an advance over previous versions, which is the key argument of the paper. The robustness of this approach must be discussed in substantial detail in any revision. Included in that discussion must be details concerning the experiments (e.g. whether the AMS measurements in Fig. 3 represent the fraction around 100 nm in the smog chamber and, if not, what the mass concentrations around 100 nm were during that experiment; what time resolution was used in the AMS measurements and how that compares to the time dependent composition changes; what is, and what factors are governing, the AMS limit of detection; what mass concentrations are required for this approach to be viable; etc.).*

Author's answer

1. The authors hope that the changes made in response to Anonymous Reviewer 1's comments will address many of the concerns regarding data quality. In short, by averaging the AMS measurements of chamber-generated SOA to a lower time resolution, both the detection limits

and the variability of most species were substantially improved. As a result, it was no longer necessary to artificially smooth the data. NH$_4$ was an exception, as its AMS signal is commonly less stable than other species and was more strongly affected by the reduced signal to noise ratio offered by PTOF measurements. Since NH$_4$ contributes a significant fraction of AS mass, it could not be excluded from the analysis, however it was necessary to reduce its variability by estimating NH$_4$ concentrations using a simple linear fit to the NH$_4$ PTOF data. The resulting composition-based HGF models still obtain close agreement with the direct VH-TDMA measurements, so we are confident that this methodology offers meaningful results even when sampling rapidly changing, size-dependent aerosol. Section 3.1 has been substantially rewritten to discuss the revised analysis and results from this experiment. We would like to also directly respond to the specific concerns mentioned in the referee's comment.

**Mass concentrations at 100 nm:**
Fig 3a (now renumbered as Fig 4) represents the bulk mass concentrations of non-refractory species. The caption and corresponding text has been reworded to clarify this. A new figure has been added to the supplement (Fig S3) which gives the heated and unheated mass concentrations for aerosol at 100 nm.

**AMS time resolution**
As mentioned on P3 L17, the combined system typically samples with a time resolution of 3 minutes per sample (6 minutes per unheated/heated cycle). This time resolution was halved for the chamber experiment to obtain satisfactory signal averaging while being fast enough to observe trends in the composition during SOA formation (discussed on P7 L19).

**AMS limit of detection**
A brief discussion of AMS detection limits and uncertainties has been added to Section 2.1 (P2, L33). Further discussion regarding the sensitivity of AMS measurements, and limitations of using PTOF measurements has been included in Section 3.1 (P7, L15),

**Required mass concentrations**
We have not attempted to estimate necessary mass concentrations for this approach, because it is situational. In essence, there needs to be a balance between the required time resolution, size resolution and available mass concentration. For example, mass concentrations were very low during the Cape Grim experiment, however meaningful observations were still possible by analyzing bulk aerosol measurements and using a low time resolution.

*Referee's comment*
*2. The second major comment relates to the use of the thermodenuder approach. It is known that many chemical components of secondary organic aerosol (e.g. oligomers) can thermally decompose when passed through a thermodenuder at temperatures as low as 100ºC (Hall and Johnston, Aerosol Sci. Technol., **2012**, 46, 983-989). This observation may have a significant impact on the interpretation of the VHTDMA measurements, especially since the thermodenuder used in this manuscript is ramped up to 500ºC. In the revised manuscript, the authors should include a discussion of the limitations of the thermodenuder approach with respect to separation of semi-volatile and low volatility components against likely changes to aerosol chemical composition resulting from thermal decomposition within the thermodenuder.*

Author's answer

2. The potential for chemical change due to heating is an important underlying consideration which affects many thermal sampling processes. We appreciate the inclusion of the study by Hall and Johnson, and have created a new section (Section 2.2) which discusses limitations of volatility-based sampling.

**Minor comments**

*Referee's comment*

*1. In their revised manuscript, the authors need to better clarify the temperature threshold that separates semi-volatile from low volatility. Is the cut-off at 120°C? This is inferred in the text (page 7, lines 23-25) but is not stated in a clear and direct manner. The authors should more clearly define what is meant (functionally) by SVOC and LVOC.*

Author's answer

1. An additional section, Section 2.2 Thermal volatility measurements has been added to clarify the volatility-based assumptions and classifications that have been used in this study.

*Referee's comment*

*2. Page 4, line 10: Do the authors mean "If the aerosol chemical composition is strongly size-dependent…."?*

Author's answer

2. This section (Section 2.3) has been significantly rewritten, and this comment has been incorporated into the new discussion.

*Referee's comment*

*3. The authors should ensure all references are accurate. For example, Cerully et al. (2017) and Huldebrandt Ruiz et al. (2015) were both published in Atmos. Chem. Phys. but their references indicate Atmos. Meas. Tech. as journal in which they were published.*

Author's answer

3. Thank you for identifying these errors. All references have been checked and corrected.

**Determining the link between hygroscopicity and composition for semi-volatile aerosol species**

Joel Alroe[1], Luke T. Cravigan[1],  Marc D. Mallet[1], Zoran D. Ristovski[1], Branka Miljevic[1], Chiemeriwo G. Osuagwu[1], and Graham R. Johnson[1]

[revised manuscript text omitted]

30 hygroscopic and compositional measurements (Timko et al., 2009).

**2.2 Thermal volatility measurements**

Desorption rates within a TD are kinetically limited, particularly at high concentrations, and aerosol may not reach thermodynamic equilibrium within the TD (Riipinen et al., 2010). This can lead to under-representations of particle volatility (An et al., 2007) and requires considerable analysis to determine effective saturation concentrations. Future studies may

35 extend this approach to quantify the volatility of each desorbed component; however, in this study, the TD has been operated at a single temperature (120 °C) and used simply for component separation. In line with this, the fraction desorbed within the TD has been referred to as the semi-volatile component, while the remainder has been termed the less-volatile component. This scheme implicitly assumes complete separation of the two components. The TD used in this study had a short residence time and no cooling stage, therefore it is possible that some residual proportion of the semi-volatile component have

40 remained (or recondensed) in the particle phase. As a further complication, several studies have suggested that heating the aerosol may promote chemical change, leading to either formation (Huffman et al., 2009; Denkenberger et al., 2007) or decomposition of oligomers (Hall and Johnston, 2012). So any change in aerosol properties after heating may be due to

removal of a semi-volatile component, temperatureheat-dependent chemical change of the less volatile component, or both factors. This is an inherent limitation of most thermal sampling processes, including the AMS, and requires consideration when examining the influence of desorbed species.

**2.2 3 Size-resolved composition**

5   The AMS is commonly used to assess the bulk composition of non-refractory aerosol smaller than 1 µm. However, when sampling aerosol with size-dependent composition, this bulk analysis becomes dominated by aerosol with larger diameter and mass. This study primarily focused on In this study, all measurements were performed on aerosol with $d_m = 100$ nm, as this is both sufficiently small for composition to have a significant influence on CCN-forming potential and large enough to be within the peak sensitivity range of the AMS. Therefore, for samples with strongly size-dependent composition, PTOF

10  AMS measurements were integrated over a range of $d_{va}$ which best represented aerosol at the preselected $d_m$ used by the VH-TDMA. If the aerosol is strongly size-dependent, the focus on ultrafine aerosol inhibits meaningful conclusions from bulk compositional analysis. In that case, more representative analysis can be obtained using PTOF size-resolved measurements, given with respect to $d_{va}$. This range was affected by particle density ($\rho_p$) and the Jayne shape factor ($S$), as given by the following calculation (DeCarlo et al., 2004): The conversion to $d_{va}$ from $d_m$, using particle density ($\rho_p$), unit density ($\rho_0$) and

15  the Jayne shape factor ($S$), can be calculated as follows (DeCarlo et al., 2004):

$$d_{va} = \frac{\rho_p}{\rho_0} S \times d_m \quad . \tag{1}$$

where $\rho_0$ is the unit density. Since $\rho_p$ and $S$ are composition-dependent, PTOF mass concentrations are integrated over a $d_{va}$ range which best represents aerosol at the preselected $d_m$ used by the VH-TDMA. For heated measurements, this PTOF $d_{va}$ range is was shifted downwards in proportion to the $d_m$ reduction observed with the VH-TDMA. AMS particle transmission

20  efficiency decreasess for diameters below $d_{va} = 100$ nm, reaching approximately 0 % transmission at 40 nm. Since the chamber-based measurements, discussed in Section 3.1, required Where PTOF data are required from this reduced sensitivity diameter range, a linear correction factor is was applied to account for transmission losses (Knote et al., 2011). While this use of PTOF data provides composition that is more directly relevant to the VH-TDMA measurements, it encompasses a reduced fraction of the overall aerosol mass concentration and does not benefit from as much signal

25  averaging as is available for bulk analysis. This is particularly exacerbated This is particularly exacerbated during periods of low mass concentration or rapid changes in aerosol properties. As demonstrated in Section 3.1, the time resolution must be carefully chosen to allow sufficient signal averaging while still capturing transient changes in the aerosol. when high time resolution is required, or during periods of low mass loading, and can result in a highly variable timeseries. In this study, we present two scenarios in which these challenges have been addressed by using nonparametric regression to smooth the

30  dataset (Section 3.1) or by significantly reducing the time resolution of the analysis (Section 3.2).

**2.4 3 Hygroscopic analysis**

The hygroscopicity of small particles is reduced by the Kelvin effect, which causes water activity ($a_w$) at the droplet/air interface to increase with particle curvature. To account for this, the measured HGFs can be re-expressed in terms of the hygroscopicity parameter ($\kappa$), using $\kappa$-Köhler theory (Petters and Kreidenweis, 2007):

35  $$\frac{RH/100}{\exp\left(\frac{4\sigma M_w}{RT\rho_w D_d HGF}\right)} = \frac{HGF^3 - 1}{HGF^3 - (1-\kappa)} \quad , \tag{2}$$

where RH is the relative humidity set in the H-SMPS, $\sigma$ is the droplet surface tension (assumed to be equivalent to pure water, $\sigma_w = 0.072$ J m$^{-2}$), $M_w$ is the molecular weight of water, $R$ is the universal gas constant, $T$ is the temperature, $\rho_w$ is

the density of water and $D_d$ is the dry particle diameter. The $\kappa$ values can then be reverted to Kelvin-corrected HGFs ($HGF_{corr}$) by setting $a_w$ equal to RH/100:

$$\frac{1}{a_w} = 1 + \frac{\kappa}{HGF_{corr}^3} \tag{3}$$

After excluding the effect of droplet curvature, the compositional influence on hygroscopicity can be investigated in detail.

5    The hygroscopicity of internally mixed aerosol is commonly estimated using the following volume-weighted model (Stokes and Robinson, 1966):

$$HGF_{total}^3 = \sum_i \varepsilon_i HGF_i^3 \ , \tag{4}$$

where $\varepsilon_i$ and $HGF_i$ are the volume fraction and independent HGF contribution of each component. If the components have substantially different volatilities, these parameters can be directly determined from VH-TDMA measurements. Otherwise
10    assumed HGF contributions are used and $\varepsilon_i$ is derived from the corresponding mass concentrations ($m_i$) and densities ($\rho_i$) of each component:

$$\varepsilon_i = \frac{m_i/\rho_i}{\sum_i m_i/\rho_i} \tag{5}$$

**2.54 Aerosol transmission efficiencies**

The two sampling lines offer different transmission efficiencies due to variations in tubing geometry, an additional solenoid
15    valve on the TD lineLine A, and losses associated with the TD itself. Diffusional losses in the TD have been reduced by omitting the cooling section. When sampling at high aerosol loading, this may cause recondensation of volatile species onto the aerosol as it cools at the outlet, however negligible recondensation is expected for most atmospheric samples (Saleh et al., 2011). The remaining differences in transmission efficiency have been quantified by examining size- and temperature-dependent changes in aerosol concentration between the two flow paths.

20    Ammonium sulfate (AS) aerosol was generated with a nebuliser (Mesa Labs, 6-jet Collison). The aerosol was dried, neutralised and sampled at three preselected sizes: $d_m$ = 50, 150 and 300 nm. The humidifier and H-SMPS were replaced with a condensation particle counter (CPC; TSI Model 3772) and a suitable bypass flow to maintain normal sampling flowrates throughout the system. From each sampling line, measurements were made with the TD at room temperature, to examine differences in tubing and solenoid valves. The TD temperature was then progressively increased in 5 °C increments
25    up to the volatilisation point of AS (180 °C at 50 nm in this system), to quantify thermophoretic losses.

Figure 2 displays the relative transmission efficiency of the TD lineLine A compared to the unheated lineLine B. For diameters of 150 nm and above, path-dependent losses of less than 5 % were observed at room temperature. This increased to over 12 % at 50 nm. In addition, transmission efficiency decreased linearly with increasing temperature, for all sizes. Based on these results, a constant correction factor has been applied to all size-resolved AMS mass concentrations in this
30    study, assuming a mean 85 % relative transmission efficiency for aerosol of $d_m \leq 100$ nm at 120 °C. This reduces bias between the two sampling lines and isolates the compositional changes caused by removal of volatile species.

**2.65 Smog chamber sampling**

A system test was conducted under controlled conditions to examine correlation between the two instruments and ensure that meaningful conclusions could be drawn. Measurements were performed using AS-seeded secondary organic aerosol (SOA),
35    generated in a temperature-controlled, 8 m³ Teflon© smog chamber. To minimise wall losses, this chamber has not been fitted with any mixing fans. During all chamber preparations, purified air was generated with a zero air generator (Aadco, Model 737-13). Type 2 laboratory-grade deionised water was used in all solutions and in humidifying the chamber.

The chamber was cleaned by first flushing it with a 1 ppm concentration of $O_3$ in purified air until particle number concentrations fell below 100 $cm^{-3}$. The UV lights were turned on for 10 minutes to promote particle formation, and the chamber was again flushed with ozone-free purified air until particle numbers decreased below 100 $cm^{-3}$. The chamber was then prepared by successively passing water vapour, gaseous nitrous acid (HONO), N-butanol and α-pinene into the chamber using a 15 L $min^{-1}$ carrier flow of purified air. Water vapour was flushed into the chamber from a heated glass flask of deionised water until a relative humidity of 50 % was achieved. The HONO was generated by adding a 0.15 M aqueous solution of $NaNO_2$ (Sigma-Aldrich) to 1.1 M sulfuric acid (Thermo Fisher Scientific) and was introduced over a 20 minute period. 1 µL of N-butanol (D9, 98 %, Sigma-Aldrich) and α-pinene (Sigma-Aldrich) were each vaporised in separate heated glass bulbs. The former acted as a tracer, allowing OH concentrations to be monitored with a Chemical Ionisation Time-of-Flight Mass Spectrometer (Barmet et al., 2012). A polydisperse distribution of AS seed particles were generated by nebulising an 0.07 M aqueous AS solution, using laboratory compressed air filtered through a high-efficiency particulate air filter. These were injected into the chamber until particle number concentrations reached $10^4$ $cm^{-3}$ with a geometric mean diameter of 94 nm. The chamber was then irradiated by twenty 160 W UV lamps and sampling was conducted over a four-hour period.

VH-TDMA measurements were performed on 100 nm aerosol with a 6-minute time resolution (3 minutes per sampling path). The TD was set to 120 °C, to target desorb organic compounds with higher volatility than AS. A nafion dryer was used to maintain the The inlet RH was maintained at 32.6 ± 0.3 % throughout the experiment and the H-SMPS was humidified humidifier was set to 90 88 %RH. Initial gas-phase concentrations within the chamber were 35 ppb α-pinene, 0.042 ppm NOx and approximately $1.5\times10^7$ molecules $cm^{-3}$ of OH, at a relative humidity of 56 %. However, due to time constraints, UV irradiation and sampling commenced immediately after aerosol injection was completed, so initially the chamber may not have been uniformly mixed. Figure 3 gives an example of the direct measurements obtained from both instruments, from the first 2.5 hours of alternating heated and unheated measurements.

**2.7 6 Remote coastal measurements**

To investigate the suitability of the combined system for atmospheric sampling, it was deployed to the Cape Grim Baseline Air Pollution Station for a two-week measurement campaign in March 2015. This remote site on the northwest coast of Tasmania, Australia, frequently receives strong westerly winds carrying marine aerosol with negligible terrestrial or anthropogenic influences. From 22:00 on 2nd March, these baseline conditions were observed for an 8-hour period, with mean particle number concentrations of 400 ± 60 $cm^{-3}$ for aerosol diameters above 10 nm. This period was accompanied by significantly increased sulfate-related mass concentration and a pronounced bimodal particle number size distribution (Fig. 3 4), consistent with cloud-processed marine aerosol (Hoppel et al., 1986). The final two hours of baseline sampling was accompanied by a pronounced decrease in sulfate- and ammonium-related signal and a decrease in particle number concentration, suggesting a change in the air mass and associated aerosol properties. For this reason, analysis has been focused on the initial 6 hours which exhibited the most consistent properties.

To account for the low aerosol concentrations, aerosol properties were averaged over this 6 hour period and the resulting mean values were used for all further analysis all measurements have been averaged over this period, rather than examining trends over a higher time resolution. Likewise, uncertainties have been determined from standard deviations in the means. The sampling inlet RH was consistently dried with a diffusion dryer to below 30 % and the humidifier was maintained at 90 %RH. and sampling Sampling was conducted using the same pre-selected diameter and, TD temperature and humidifier RH as in the chamber-based experiment (i.e. 100 nm, 120 °C, 90 % respectively).

**3 Results and validation**

**3.1 Chamber-generated aerosol**

A characteristic example of the alternating heated and unheated measurements obtained from both instruments is given in Fig. 4, showing the impact of an increasing SOA component on bulk aerosol composition, volatility and water uptake during the smog chamber experiment. Measurements commenced immediately after injecting the AS seeds and there was no active mixing mechanism within the chamber. In light of this, the initial rapid decrease in $SO_4$ concentrations suggest that the chamber was  well mixed during  the first 40 minutes of sampling. After this point, the total mass loading, and mode diameter of the number size distributions, progressively increased in response to SOA formation (Figs S1 and S2).

Compositional measurements of the seeded SOA indicated a strong size dependence, with the AS mass distributed around a mean $d_{va}$ of approximately 500 nm, while the SOA component progressively dominated at 100-200 nm (Fig. 5a). Since the size-distribution of each species did not change substantially over time, further compositional analysis was restricted to diameters in the range: $130 < d_{va} < 180$ nm.  This range was  selected to reflect varying proportions of internally mixed SOA and AS with densities of 1.3 and 1.78 g cm$^{-3}$, respectively (Chen and Hopke, 2009).

While these size limits ensured that the compositional analysis was directly relevant to the $d_m$ used by the VH-TDMA, it reduced the sensitivity of the AMS measurements in two ways. Firstly, this diameter range represented a comparatively low proportion of the total particle mass, decreasing the signal to noise ratio of the measurements. Secondly, by using only a subset of PTOF measurements rather than the bulk aerosol measurements, less signal averaging was applied to each sample, increasing the variability of the measurements. To account for this, the size-resolved composition was averaged to a 12 minute time resolution and the resulting time series has been provided in the supplement (Fig. S3). This resolution was sufficiently fast to observe the SOA fraction progressively growing to dominate the mass composition at 100 nm, while  ensuring that the compositional measurements were sufficiently stable to be useful for hygroscopic predictions.

At this time resolution, the estimated detection limits were 0.123, 0.012, 0.023, 0.245 and 0.023 µg m$^{-3}$ for organics, $NO_3$, $SO_4$, $NH_4$ and Chl respectively. While these values are considerably higher than when using bulk aerosol measurements, a large proportion of the PTOF measurements remained above these limits, with two notable exceptions. Chl concentrations were consistently negligible and since there was no source of this species, it has been disregarded in the subsequent analysis. In addition, the $NH_4$ background signal increased by an order of magnitude when averaged over the narrow PTOF range. $NH_4$ often exhibits higher variability than most other species. Also, immediately prior to measuring the background signal, the AMS had been sampling from the chamber during injection of the AS seeds and so some residual AS may have remained in the instrument during the filtered  background measurements . Ultimately the cause of this variable signal has not been established. In any case, $NH_4$ represented a significant fraction of the injected AS aerosol mass and therefore it was not feasible to exclude its contribution. Instead, the $NH_4$ mass concentrations were approximated with a linear fit to the $NH_4$ PTOF measurements.

[revised manuscript text omitted]

Two  measurement campaigns have been discussed which presented distinctly different and challenging sampling conditions. The capacity of both instruments to perform size-resolved measurements allowed size-dependencies to be

identified and targeted in the analysis of the chamber-based campaign. However this requirement for size-dependent composition restricted analysis to a small subset of the total aerosol mass. In addition, the rapid SOA formation required a reasonably high time resolution to capture the progressive impact on composition and hygroscopicity. Although these constraints led to limited signal averaging and elevated detection limits, the independent composition-based HGF estimates demonstrated consistent agreement with direct VH-TDMA measurements. The O:C ratio of the LVOA offered a reasonable estimate of its hygroscopic contribution. However, direct VH-TDMA measurements indicated that the HGF contribution of the SVOA component was markedly lower, despite having a very similar O:C ratio to LVOA. This may further support the conclusion that these parameters are not well correlated, as reported by Hildebrandt Ruiz et al. (2015), and demonstrates the benefit of the complementary hygroscopic, volatility and compositional measurements  offered by the combined sampling system.

In contrast to the chamber experiment, aerosol number and mass concentrations were very low during the marine atmospheric measurements, but by averaging across the full 6-hour period and using bulk compositional measurements, the two instruments obtained meaningful measurements consistent with characteristic marine aerosol properties. The distribution of HGFs in the marine aerosol revealed an external mixture of nss sulfates and SSA. From these, an internally mixed semi-volatile fraction was separated and attributed to sulfuric acid and an OA component with a low degree of oxidation (O:C = 0.24). Finally, hygroscopic modelling supported the assumption that this semi-volatile component was common to both aerosol types.

[revised manuscript text omitted]

[a] **(Ault et al., 2013; Hersey et al., 2009)**

[b] **(Washburn, 1926)**

[c] **(Xiong et al., 1998)**

**Table 2: Hygroscopicity of volatility-resolved fractions observed in externally mixed marine aerosol**

| Classification | Semi-volatile volume fraction | Semi-volatile HGF | Heated HGF | Total unheated HGF |
|---|---|---|---|---|
| nss sulfate | $0.12 \pm 0.02$ | $1.2 \pm 0.3^{a}$ | $1.61 \pm 0.02$ | $1.57 \pm 0.01$ |
| SSA | $0.12 \pm 0.02$ | $1.2 \pm 0.3^{a}$ | $2.01 \pm 0.05$ | $1.90 \pm 0.04$ |

[a] **Derived from VH-TDMA measurements of nss sulfate aerosol**